# A Unified Theory of Stochastic Proximal Point Methods without Smoothness

## Abstract

This paper presents a comprehensive analysis of a broad range of variations of the stochastic proximal point method (SPPM). Proximal point methods have attracted considerable interest owing to their numerical stability and robustness against imperfect tuning, a trait not shared by the dominant stochastic gradient descent (SGD) algorithm. A framework of assumptions that we introduce encompasses methods employing techniques such as variance reduction and arbitrary sampling. A cornerstone of our general theoretical approach is a parametric assumption on the iterates, correction and control vectors. We establish a single theorem that ensures linear convergence under this assumption and the $\mu$-strong convexity of the loss function, and without the need to invoke smoothness. This integral theorem reinstates best known complexity and convergence guarantees for several existing methods which demonstrate the robustness of our approach. We expand our study by developing three new variants of SPPM, and through numerical experiments we elucidate various properties inherent to them.

## 1 Introduction

In this paper we consider the stochastic optimization problem

$$\min_{x \in \mathbb{R}^d} \left\{ f(x) := \mathrm{E}_{\xi \sim \mathcal{D}} \left[ f_\xi(x) \right] \right\}, \tag{1}$$

where $\xi \sim \mathcal{D}$ is a random variable following distribution $\mathcal{D}$, and $\mathrm{E}\left[\cdot\right]$ denotes mathematical expectation. Problems of this type are fundamental for statistical supervised learning theory. Here, $x$ is a machine learning model of $d \in \mathbb{N}$ parameters, $\mathcal{D}$ is an unknown distribution of labeled examples, samples $\xi \sim \mathcal{D}$ are available, $f_\xi$ is the loss on datapoint $\xi$, $f$ is the generalization error. In such a setup, an unbiased estimator of the gradient $\nabla f_\xi(x)$ is computed instead of the gradient $\nabla f(x)$. We rely on two assumptions, presented next.

**Assumption 1.** *Function $f_\xi : \mathbb{R}^d \to \mathbb{R}$ is differentiable for all samples $\xi \sim \mathcal{D}$.*

We implicitly assume that the order of differentiation and expectation can be swapped, which means that $\nabla f(x) \overset{(1)}{=} \nabla \mathrm{E}_{\xi \sim \mathcal{D}} \left[ f_\xi(x) \right] = \mathrm{E}_{\xi \sim \mathcal{D}} \left[ \nabla f_\xi(x) \right].$ This implies that $f$ is differentiable.

**Assumption 2.** *Function $f_\xi : \mathbb{R}^d \to \mathbb{R}$ is $\mu$-strongly convex for all samples $\xi \sim \mathcal{D}$, where $\mu > 0$ :*

$$f_\xi(y) + \langle \nabla f_\xi(y), x - y \rangle + \frac{\mu}{2} \|x - y\|^2 \le f_\xi(x), \quad \forall x, y \in \mathbb{R}^d. \tag{2}$$

This implies that $f$ is $\mu$-strongly convex, and hence $f$ has a unique minimizer, which we denote by $x_\star$. We know that $\nabla f(x_\star) = 0$. Notably, we do *not* assume $f$ to be $L$-smooth.

Another type of problem considered in the paper is a minimization of functions $f$ that are averages of a large number of differentiable functions:

$$\min_{x \in \mathbb{R}^d} \left\{ f(x) := \frac{1}{n} \sum_{i=1}^{n} f_i(x) \right\}. \tag{3}$$

This is a special case of (1), with $\mathcal{D}$ being the uniform distribution over the finite set $[n]$: $\xi = i$ with probability $\frac{1}{n}$. Problems with this structure commonly emerge in practice during the training of supervised machine learning models via empirical risk minimization. They are known as finite-sum optimization problems. We rely on two assumptions, presented next.

**Assumption 3.** *Function $f_i : \mathbb{R}^d \to \mathbb{R}$ is differentiable for all $i \in [n]$.*

This implies that $f$ is differentiable.

**Assumption 4.** *Function $f_i : \mathbb{R}^d \to \mathbb{R}$ is $\mu_i$-strongly convex for all $i \in [n]$, where $\mu_i > 0$. That is,*

$$f_i(y) + \langle \nabla f_i(y), x - y \rangle + \frac{\mu_i}{2} \|x - y\|^2 \leq f_i(x), \quad x, y \in \mathbb{R}^d. \tag{4}$$

This refines Assumption 2, with the the same strong convexity parameter for all functions. This implies that $f$ is $\mu$-strongly convex with $\mu = \min_i \mu_i$. Hence, $f$ has a unique minimizer $x_\star$.

## 2 Variations of SPPM

The most common algorithm for finding an $\varepsilon$-accurate solution of problems (1) and (3) is stochastic gradient descent (SGD)Robbins and Monro (1951); Nemirovski et al. (2009); Gower et al. (2019). In general, SGD updates have the form of $x^{k+1} = x^k - \gamma g^k$, where $g^k$ is an unbiased stochastic gradient estimator: $\mathrm{E}\left[g^k \middle| x^k\right] = \nabla f(x^k)$. There are numerous approaches to constructing an estimator with the aim of attaining preferable algorithmic features such as faster convergence rate, lower iteration cost, parallelization, and generalization. One of the most significant issues in SGD variations is the difficulties with a suitable selection of the stepsize. Theoretical results highlight that the stepsize is restricted to small values (Bach and Moulines, 2011).

Whenever we are able to evaluate the stochastic proximity operator, another option to consider in place of SGD is the stochastic proximal point method (SPPM), whose iterations have the form

$$\mathrm{prox}_{\gamma f_{\xi_k}}(x_k) := \arg \min_{x \in \mathbb{R}^d} \left\{ f_{\xi_k}(x) + \frac{1}{2\gamma} \|x - x_k\|^2 \right\}, \quad \xi_k \sim \mathcal{D},$$

where $\gamma > 0$ is a stepsize. Clearly, the proximity operator is well-defined due to the strong convexity Assumption 2 on $f_\xi(x)$. Given the error tolerance $\varepsilon > 0$, we derive in Appendix B.2 (see (23); also, see (Asi and Duchi, 2019, Proposition 5.3), (Khaled and Jin, 2023, Theorem 1)) that with $\sigma_\star^2 := \mathrm{E}_{\xi \sim \mathcal{D}} \left[ \|\nabla f_\xi(x_\star)\|^2 \right]$, the stepsize $\gamma = \frac{\mu \varepsilon}{\sigma_\star^2}$, we get $\mathrm{E}\left[ \|x_k - x_\star\|^2 \right] \leq \varepsilon$ provided that

$$k \geq \left( \frac{1}{2} + \frac{\sigma_\star^2}{2\mu^2 \varepsilon} \right) \log \left( \frac{2\|x_0 - x_\star\|^2}{\varepsilon} \right).$$

As shown by Gower et al. (2019), in the same setting, SGD with fixed stepsize reaches an $\varepsilon$-accurate solution after

$$k \geq \left( \frac{2L}{\mu} + \frac{2\sigma_\star^2}{\mu^2 \varepsilon} \right) \log \left( \frac{4\|x_0 - x_\star\|^2}{\varepsilon} \right)$$

iterations ($L$ is a bound on the smoothness constant of stochastic functions). Note that although both iteration complexities are dependent on the stochastic noise term, the iteration complexity of SGD additionally hinges on the condition number. $\kappa := \frac{L}{\mu} \geq 1$ ($\mu$ is a strong convexity parameter of $f$ here). In contrast, the iteration complexity of SPPM remains unaffected by the smoothness constant $L$. Consequently, if we have access to stochastic proximal operator evaluations, we can achieve a faster convergence rate than SGD. Another important aspect is that SPPM still works for large stepsizes. *The primary distinction* with the result of Khaled and Jin (2023) is that the neighborhood guaranteed in our analysis for SPPM does not blow up to infinity as the stepsize $\gamma$ grows to infinity (see (21) in Theorem 2 and Commentary 2 after it).

Ryu and Boyd (2016); Asi and Duchi (2019) demonstrated that SPPM exhibits greater resilience in terms of the choice of stepsize compared to SGD. Ryu and Boyd (2016) furnish convergence rates for SPPM and note its stability against learning rate misspecification, a trait not shared by SGD. Asi and Duchi (2019) examine a broader method (AProx) which encompasses SPPM as a particular instance, providing both stability and convergence rates under convexity. Additionally, the convergence rates of SPPM remain consistent with those of SGD across different versions of the algorithms.

We have discussed the versatility in designing the unbiased stochastic estimator $g^k$ in SGD, which can be accomplished in various manners. Among these, popular sampling strategies include importance sampling and mini-batching. A comprehensive analysis of these methods within the framework of arbitrary sampling was presented by Gower et al. (2019). Similarly, an analogous effort is

made for SPPM in our paper, where sampling schemes and methods such as SPPM-NS and SPPM-AS are proposed (see Appendix B.3 and B.4). While sampling strategies offer significant utility, both for SGD and SPPM variations, they tend to converge to the vicinity of the solution for fixed stepsizes, reaching the exact solution only in the overparameterized regime.

That being said, the issue of SGD iterates not converging to the optimum prompted the development of variance-reduced methods. These methods enhance the convergence rate of SGD for finite-sum problems by constructing gradient estimators with diminishing variance over time (e.g., SAGA (Defazio et al., 2014) and SVRG (Johnson and Zhang, 2013)). In other words, variance-reduced methods progressively acquire knowledge of the stochastic gradients at the optimum and mitigate the influence of gradient noise. Consequently, for strongly convex $f$, these variance-reduced methods exhibit linear convergence towards $x_\star$ with a fixed step size (see a survey by Gower et al. (2020)). In the discussion above we concluded that SPPM has advantages over SGD. This motivated Khaled and Jin (2023) to explore variance-reduced variants of SPPM. They proposed the SVRP algorithm with an approximate calculation of the proximal operator and demonstrated that variance reduced variants of SPPM have better convergence guarantees under second-order similarity assumption for the finite-sum setting (3). We present and employ in this paper a more general form of this assumption for the stochastic setting (1) in order to analyze L-SVRP with the proximal operator in the updates. We stipulate the similarity assumption to be valid exclusively at $x_\star$, whereas Khaled and Jin (2023) require it to hold for any $y \in \mathbb{R}^d$ instead.

**Assumption 5** (Similarity). *There exists $\delta \geq 0$ such that*

$$\mathrm{E}_{\xi \sim \mathcal{D}} \left[ \|\nabla f_\xi(x) - \nabla f(x) - \nabla f_\xi(x_\star)\|^2 \right] \leq \delta^2 \|x - x_\star\|^2, \qquad \forall x \in \mathbb{R}^d. \tag{5}$$

Let us also note here that the standard $\delta$-smoothness assumption implies the second-order similarity assumption by Khaled and Jin (2023), which in turn implies Assumption 5. In Appendix B.6 we discuss the generality of Assumption 5 in detail. In the studies of Szlendak et al. (2022); Panferov et al. (2024), federated optimization with compression under the similarity assumption is explored, leading to improved convergence rates achieved through specially designed compression operators.

Point SAGA, also a variant of SPPM, was proposed by Defazio (2016), where its convergence was analyzed under the individual smoothness assumption. The algorithm requires a large amount of memory for its execution. It inherits this shortcoming from its SGD variance-reduced archetype SAGA by Defazio et al. (2014). The structure of Point SAGA will not allow us to perform the analysis under the similarity assumption (Assumption 5). Instead, we will rely on this stronger assumption.

**Assumption 6.** *We assume that there exists $\nu > 0$ such that the inequality*

$$\frac{1}{n} \sum_{j=1}^{n} \left\| \nabla f_j(x^j) - \frac{1}{n} \sum_{i=1}^{n} \nabla f_i(x^i) - \nabla f_j(x_\star) \right\|^2 \leq \nu^2 \frac{1}{n} \sum_{j=1}^{n} \|x^j - x_\star\|^2 \tag{6}$$

*holds for all $x^1, \ldots, x^n \in \mathbb{R}^d$.*

Let us note that this condition is weaker than the individual $\nu$-smoothness assumption. In Appendix B.8 we discuss the generality of Assumption 6 in more detail.

Traoré et al. (2023) analyze L-SVRP and Point SAGA algorithms in the setting (3) when each $f_i$ is smooth, convex, and either $f$ is convex or satisfies the PŁ-condition. In contrast, in our work we consider a different setting with individually strongly convex functions under Assumption 5 or Assumption 6 (both are weaker than the individual smoothness assumption). The convergence theory of variance-reduced SPPM methods significantly differs from that of standard SPPM. We suggest the possibility of a unified theory that encompasses both SPPM and its variance-reduced counterparts.

## 3 CONTRIBUTIONS

Numerous efficient adaptations of the SPPM algorithm have emerged, each with its specific applications. Our research stems from the absence of a comprehensive and universally applicable theory. While some connections among existing methods have been established (as demonstrated

by Traoré et al. (2023), who link the classical SPPM method with its variance-reduced counterparts), a cohesive theoretical framework without the smoothness assumption is missing. Understanding the connections among various algorithms rooted in SPPM is becoming increasingly challenging for the community, both in theory and practical applications. While new variants are yet to be discovered, determining concrete principles beyond intuition to guide their discovery remains challenging. Complicating matters further is the use of various assumptions regarding the correction vectors across different literature, each with differing levels of rigor. Key contributions of this study comprise:

• **A universal algorithm.** We design a universal SPPM-LC algorithm (stochastic proximal point method with learned correction, Algorithm 1) that encompasses 7 variants of stochastic proximal point method (SPPM, SPPM-NS, SPPM-AS, SPPM*, SPPM-GC, L-SVRP and Point-SAGA; see Table 1) through *(random) correction vectors* $h_k$. A specific choice of correction vectors allows to retrieve any particular of the mentioned algorithms. Motivated by the work of Gorbunov et al. (2020), a similar universal algorithm was proposed by Traoré et al. (2023), but under different assumptions.

• **Comprehensive analysis.** We introduce a cohesive theoretical framework of assumptions that includes three restrictions imposed on correction vectors $h_k$ of the SPPM-LC algorithm (Assumption 7). These restrictions connect the correction vectors $h_k$ to the iterates of the algorithm and have the forms of parametric recursions. The development of this framework constitutes one of our major contributions. Under Assumption 1, Assumption 2 on the functions and Assumption 7 on the correction vectors, we analyze the convergence of SPPM-LC in the case when it is applied to find an $\varepsilon$-accurate solution of the general stochastic optimization problem (1). This is the first comprehensive analysis of the original, sampling-based, and variance-reduced versions of SPPM. It implies the convergence results for all 7 variations of stochastic proximal point method (SPPM, SPPM-NS, SPPM-AS, SPPM*, SPPM-GC, L-SVRP and Point-SAGA). The sampling-based methods SPPM-NS and SPPM-AS are employed for a less general setting (3), as well as the variance-reduced method Point SAGA, they are analyzed under Assumption 3 and Assumption 4 on the functions. In addition, the analysis of L-SVRP is based on the similarity Assumption 5, and the analysis of Point-SAGA is based on Assumption 6. In order to obtain the convergence result for any particular of the mentioned algorithms, one needs to check that (the recursive parametric) Assumption 7 holds for it. Traoré et al. (2023) introduce a slightly different theoretical framework of assumptions and analyze SPPM and its variance-reduced versions in a finite-sum setting (3) in another setup when functions are individually convex and $L$-smooth.

• **Best known rates retrieved.** The rates derived from our comprehensive Theorem 1, under Assumption 1, Assumption 2 on the functions for SPPM, SPPM*, SPPM-GC, L-SVRP (under additional Assumption 5) and Point-SAGA (under additional Assumption 6); under Assumption 3 and Assumption 4 on the functions for SPPM-NS, SPPM-AS, represent the sharpest rates for these methods. Notably, the neighborhood that we guarantee for SPPM in our analysis does not blow up for the stepsize $\gamma \to \infty$ in comparison to the result of Khaled and Jin (2023) (see (21) in Theorem 2 and Commentary 2 after it). Also, we present the analysis for L-SVRP in a simplified way and obtain slightly better bounds on the iteration complexity up to a constant factor than Khaled and Jin (2023).

• **Analysis under general similarity assumptions.** The analysis of L-SVRP is based on the similarity Assumption 5, and the analysis of Point-SAGA is based on Assumption 6. Both of these assumptions are very general and distinguish our approach from the previous ones. Defazio (2016) analyzed Point SAGA under the individual smoothness assumption, which implies Assumption 6. Khaled and Jin (2023) analyzed L-SVRP under a more restrictive similarity assumption than Assumption 5. Traoré et al. (2023) analyzed L-SVRP and Point SAGA under the individual smoothness assumption, which is much more restrictive than our Assumption 5 and Assumption 6. Their results were obtained prior to our work, but we conducted our analysis independently: we already had our result when we discovered their paper.

• **Analysis in a general setting.** We analyze L-SVRP in the stochastic optimization setting (1) while previous works of Khaled and Jin (2023); Traoré et al. (2023) do this in the less general setting (3).

• **New methods.** Our comprehensive theory offers complexity bounds for a range of novel (SPPM-LC, SPPM-NS, SPPM-AS, SPPM-GC, SPPM*) and upcoming variations of SPPM. It suffices to confirm that Assumption 7 holds, and a complexity estimate is readily provided by Theorem 1. Selected existing and new methods that align with our framework are outlined in Table 1.

• **Experiments.** Through extensive experimentation, we demonstrate that several of the newly introduced methods, analyzed within our framework, exhibit compelling empirical properties when compared to natural baselines.

## 4 MAIN RESULT

We are now ready to present our general Algorithm 1, which we call Stochastic Proximal Point Method with Learned Correction (SPPM-LC). Subsequently, we introduce the core assumption on the correction vectors, iterates and control vectors of Algorithm 1 enabling our general analysis, and further state and comment on our unified convergence result (Theorem 1).

---

**Algorithm 1** Stochastic Proximal Point Method with Learned Correction (SPPM-LC)

---

1: **Parameters:** learning rate $\gamma > 0$, starting point $x_0 \in \mathbb{R}^d$, control vector $\phi_0 \in \mathbb{R}^m$
2: **for** $k = 0, 1, 2, \ldots$ **do**
3:     Sample $\xi_k \sim \mathcal{D}$
4:     Form correction vector $h_k$ as a function of the iterate $x_k$, control vector $\phi_k$, and sample $\xi_k$
5:     $x_{k+1} = \mathrm{prox}_{\gamma f_{\xi_k}} (x_k + \gamma h_k)$
6:     Construct a new control vector $\phi_{k+1}$
7: **end for**

---

### 4.1 KEY ASSUMPTION

We assume that the (random) correction vector $h_k$ has zero mean (conditioned on $x_k$ and $\phi_k$, the $k$-th iterate and control vector, respectively), and that it is connected with the iterates of SPPM-LC via two parametric recursions/inequalities, described next. We introduce versatility by expressing these inequalities parametrically.

**Assumption 7** (Parametric recursions). *Let $\{x_k, \phi_k\}_{k \geq 0}$ be the random iterates produced by SPPM-LC. Assume that*

$$\mathrm{E}\left[ h_k \,\middle|\, x_k, \phi_k \right] = 0. \tag{7}$$

*Further, assume that there exists a function $\sigma^2 : \mathbb{R}^m \to \mathbb{R}_+$ and nonnegative constants $A_1, B_1, C_1, A_2, B_2, C_2$, with $B_2 < 1$, such that*

$$\mathrm{E}\left[ \|h_k - \nabla f_{\xi_k}(x_\star)\|^2 \,\middle|\, x_k, \phi_k \right] \;\leq\; A_1\|x_k - x_\star\|^2 + B_1\sigma_k^2 + C_1, \tag{8}$$

$$\mathrm{E}\left[ \sigma_{k+1}^2 \,\middle|\, x_{k+1}, \phi_k \right] \;\leq\; A_2\|x_{k+1} - x_\star\|^2 + B_2\sigma_k^2 + C_2, \tag{9}$$

*where $\sigma_k^2 := \sigma^2(\phi_k)$.*

For brevity, we refer to this assumption as the "$\sigma_k^2$ assumption". If Assumption 7 holds, then by taking expectation on both sides of (8) and (9) and applying the tower property in each case, we get

$$\mathrm{E}\left[ \|h_k - \nabla f_{\xi_k}(x_\star)\|^2 \right] \;\leq\; A_1\mathrm{E}\left[ \|x_k - x_\star\|^2 \right] + B_1\mathrm{E}\left[ \sigma_k^2 \right] + C_1, \tag{10}$$

$$\mathrm{E}\left[ \sigma_{k+1}^2 \right] \;\leq\; A_2\mathrm{E}\left[ \|x_{k+1} - x_\star\|^2 \right] + B_2\mathrm{E}\left[ \sigma_k^2 \right] + C_2, \tag{11}$$

The novelty of our approach lies in the introduction of inequalities (8) and (9). We support and validate this assertion by providing numerous examples (in Section 5), demonstrating that these inequalities encapsulate the nature of a broad range of existing SPPM methods as well as some new ones, including standard SPPM alongside its arbitrary sampling and variance-reduced variants. In its essence, we generalize, parameterize and establish as an independent assumption the conditions on correction vectors for SPPM-type methods present in the literature, regardless of the specifics defining the base method from which they stem. Traoré et al. (2023) propose different parameterized recursive inequalities and analyze SPPM and its variance-reduced versions in a finite-sum setting (3) under the condition where functions are individually convex and $L$-smooth. Similar inequalities can be found in the analysis of SGD-type methods (a unified theory developed by Gorbunov et al. (2020)).

Table 1: Compilation of both existing and novel methods that align with our comprehensive analytical framework. Problem (1) encompasses a broader scope compared to problem (3). VR = variance reduced method, AS = arbitrary sampling. Thm $x$ = Thm 1 + Lemma $x$, Lemma $x$ is in Section A.$x$, $x \in [8]$. The last column indicates whether the analysis is new or not and whether we recover the previously established rate.

| Problem | Method | Alg # | VR? | AS? | Section | Result / Rate |
|---------|--------|-------|-----|-----|---------|---------------|
| (1) | SPPM-LC **[NEW]** | Alg 1 | ✓✗ | ✗ | B.1 | Thm 1 **[NEW]** |
| (1) | SPPM [1] (Bertsekas, 2011) | Alg 2 | ✗ | ✗ | B.2 | Thm 2 **[NEW]** |
| (3) | SPPM-NS **[NEW]** | Alg 3 | ✗ | ✗ | B.3 | Thm 3 **[NEW]** |
| (3) | SPPM-AS **[NEW]** | Alg 4 | ✗ | ✓ | B.4 | Thm 4 **[NEW]** |
| (1) | SPPM* **[NEW]** | Alg 5 | ✓ | ✓ | B.5 | Thm 5 **[NEW]** |
| (1) | SPPM-GC **[NEW]** | Alg 6 | ✓ | ✗ | B.6 | Thm 6 **[NEW]** |
| (1) | L-SVRP [2] (Khaled and Jin, 2023) | Alg 7 | ✓ | ✗ | B.7 | Thm 7 **[NEW]** |
| (3) | Point SAGA [3] (Defazio, 2016) | Alg 8 | ✓ | ✗ | B.8 | Thm 9 **[NEW]** |

[1] SPPM was studied by Khaled and Jin (2023) under the less general similarity assumption than ours. Bertsekas (2011) proposed incremental proximal point method (SPPM for problem (3)) and analyzed it under assumptions that each $f_i$ is Lipschitz. We guarantee in our analysis that the neighborhood does not blow up when the stepsize is large and consider the general setting (1).

[2] The L-SVRP method was proposed by Khaled and Jin (2023) and called SVRP therein. It was inspired by the L-SVRG method of Hofmann et al. (2015); Kovalev et al. (2020), who were in turn inspired by the SVRG method of Johnson and Zhang (2013). Following Khaled and Jin (2023), we use the name L-SVRP to highlight the loopless nature of the update of control vector. The theoretical results presented here are a minor adaptation of the results of Khaled and Jin (2023). We present the analysis in a simplified way, and hence obtain slightly better bounds up to a constant factor. Khaled and Jin (2023) employ an approximation of the proximal operator in the updates of L-SVRP while we use the operator itself.

[3] Point SAGA was proposed by Defazio (2016). The main difference between our form and the original one is in the control vectors. Defazio (2016) updates a table with gradients, while we update a table with points at which we compute the gradients.

## 4.2 MAIN THEOREM

We are now prepared to introduce our main convergence result.

**Theorem 1.** *Let Assumption 1 (differentiability) and Assumption 2 ($\mu$-strong convexity) hold. Let $\{x_k, h_k\}$ be the iterates produced by SPPM-LC (Algorithm 1), and assume that they satisfy Assumption 7 ($\sigma_k^2$-assumption). Choose any $\gamma > 0$ and $\alpha > 0$ satisfying the inequalities*

$$\frac{(1 + \gamma^2 A_1)(1 + \alpha A_2)}{(1 + \gamma\mu)^2} < 1, \qquad \frac{\gamma^2 B_1(1 + \alpha A_2)}{\alpha(1 + \gamma\mu)^2} + B_2 < 1, \tag{12}$$

*and define the Lyapunov function*

$$\Psi_k := \|x_k - x_\star\|^2 + \alpha\sigma_k^2. \tag{13}$$

*Then for all iterates $k \geq 0$ of SPPM-LC we have*

$$\mathrm{E}\left[\Psi_k\right] \leq \theta^k \Psi_0 + \frac{\zeta}{1 - \theta}, \tag{14}$$

*where the parameters $0 \leq \theta < 1$ and $\zeta \geq 0$ are given by*

$$\theta = \max\left\{\frac{(1 + \gamma^2 A_1)(1 + \alpha A_2)}{(1 + \gamma\mu)^2}, \frac{\gamma^2 B_1(1 + \alpha A_2)}{\alpha(1 + \gamma\mu)^2} + B_2\right\}, \tag{15}$$

$$\zeta = \frac{\gamma^2 C_1(1 + \alpha A_2)}{(1 + \gamma\mu)^2} + \alpha C_2. \tag{16}$$

Theorem 1 proves a linear rate of convergence for a number of stochastic proximal point methods towards a fluctuation neighborhood around the solution, regulated by the additive term in (14). It depends on parameters $C_1$ and $C_2$. The neighborhood vanishes (i.e., $\zeta = 0$) iff $C_1 = C_2 = 0$. If this happens, then SPPM-LC converges linearly to the solution as $k \to \infty$ for any fixed $\gamma > 0$, satisfying the conditions of Theorem 1. Notice that $\theta \geq B_2$, and hence the linear rate can not be faster than $(B_2)^k$. That is, as shown in Appendix B (also, see Table 2), the main difference between variance-reduced versions of SPPM and its other variants is that the former methods satisfy $\sigma_k^2$-assumption with $C_1 = C_2 = 0$ (and reach the optimum $x_\star$), whilst the latter have either $C_1 > 0$ or $C_2 > 0$.

## 5 OVERVIEW OF SPECIFIC METHODS AND OF THE FRAMEWORK

In this section we demonstrate the descriptive power of our new framework. We show that most popular existing methods can be expressed in terms of our framework and, in addition to that, we describe several new methods in more detail (see Table 1).

### 5.1 A BRIEF OVERVIEW

As asserted, the proposed framework is powerful enough to include methods without variance reduction (✗ in the "VR" column) alongside variance-reduced methods (✓ in the "VR" column), methods that fall under the arbitrary sampling paradigm (✓ in the "AS" column). All novel methods introduced in this paper are clearly designated with the label **[NEW]**. Additionally, to facilitate a thorough understanding of all algorithms discussed, detailed explanations are included in the Appendix. A link is provided for convenient navigation to the supplementary section. The generality of our framework is reflected in Table 1. The "Result / Rate" column of Table 1 refers to a Theorem $x$ which follows from Theorem 1 and Lemma $x$, $x \in [8]$. The convergence results of the algorithms considered in the paper are outlined in these theorems, offering insights into their performance characteristics. Importantly, in instances where established methods are recovered, the best-known convergence rates are reaffirmed or better results are obtained. This underscores the robustness and reliability of our analytical framework in accurately capturing the behavior of established algorithms.

### 5.2 PARAMETERS OF THE FRAMEWORK

The algorithms in Table 1 demonstrate specific patterns in relation to the parameters in Assumption 7. To elucidate this observation, we provide a summary of these parameters in Table 2. As predicted by Theorem 1, when $C_1 = C_2 = 0$, the corresponding method does not oscillate and converges to the optimum $x_\star$, indicating the variance-reduced nature of the algorithm. All parameters referenced in Table 2 are defined in the Appendix, alongside descriptions and analyses of the respective methods.

### 5.3 FIVE NOVEL ALGORITHMS

To showcase the efficiency of our comprehensive framework, we develop three new variants of SPPM which have not previously been addressed in the literature (see Table 1). In this section, we briefly outline the reasoning behind their implementation. Further specifics are available in the Appendix.

SPPM-NS (Algorithm 3). The method is designed for solving the problem (3). Let positive numbers $p_1, \dots, p_n$ sum up to 1, set $i_k = i$ with probability $p_i$, $i \in [n]$. The step of the method has the form $x_{k+1} = \text{prox}_{\frac{\gamma}{np_i} f_{i_k}}(x_k)$. It unifies several powerful sampling strategies (as, e.g., importance sampling). Sampling allows to improve the convergence rate and modify the neighborhood.

SPPM-AS (Algorithm 4). The method is also designed for the problem (3). The arbitrary sampling framework was developed for SGD by Gower et al. (2019). It allows to conduct a sharp unified convergence analysis for various effective sampling and mini-batch strategies. For strongly convex functions, the method with constant stepsize converges linearly to the neighborhood of the solution.

SPPM* (Algorithm 5). This novel algorithm links conventional and variance-reduced SPPM methods. Although not immediately practical, it offers valuable insights into the inner workings of variance reduction. This method addresses the fundamental question: assuming that the gradients $\nabla f_\xi(x_\star)$ are available, can they be leveraged to devise a more potent variant of SPPM? The affirmative answer culminates in the development of SPPM*. The construction of updates in SPPM* involves correction vectors of the form $h_k = \nabla f_{\xi_k}(x_\star)$. In essence, this implies augmenting $x_k$ with gradients of the same functions at the optimal point $x_\star$, with respect to which the proximal operator is calculated. As evidenced in Table 2, where $C_1 = C_2 = 0$, this method converges directly to $x_\star$ without oscillation, rather than converging to a neighborhood of the solution. Practical variance-reduced methods operate by iteratively refining estimates of $\nabla f_{\xi_k}(x_\star)$. Notably, the term $\sigma_k^2$ in the Lyapunov function of variance-reduced methods converges to zero.

SPPM-GC (Algorithm 6). This method can be viewed as a practical variant of SPPM* featuring a computable version of the correction vector, $h_k = \nabla f_{\xi_k}(x_k) - \nabla f(x_k)$, instead of the incomputable correction $\nabla f_{\xi_k}(x_\star)$. This method follows the same paradigm of iteratively constructing increasingly more refined estimates of $\nabla f_{\xi_k}(x_\star)$ and is also variance reduced. As indicated in Table 2, where $C_1 = C_2 = 0$, this method also converges to $x_\star$, and not to some neighborhood of the solution only.

SPPM-LC (Algorithm 1). This new generic algorithm constructs the updates with correction vectors $h_k$ of a universal form. We analyze its behavior under a new parametric Assumption 7. The convergence result then follows. This is a unified analysis of all stochastic proximal point methods we have encountered so far: SPPM, SPPM-NS, SPPM-AS, SPPM*, SPPM-GC, L-SVRP and Point-SAGA. One only needs to check that this parametric assumption holds in each particular case (see Table 2).

| Method | $A_1$ | $B_1$ | $C_1$ | $A_2$ | $B_2$ | $C_2$ | Lemma |
|--------|-------|-------|-------|-------|-------|-------|-------|
| SPPM-LC | $A_1$ | $B_1$ | $C_1$ | $A_2$ | $B_2$ | $C_2$ | Lemma 1 |
| SPPM | 0 | 0 | $\sigma_\star^2$ | 0 | 0 | 0 | Lemma 2 |
| SPPM-NS | 0 | 0 | $\sigma_{\star,\mathrm{NS}}^2$ | 0 | 0 | 0 | Lemma 3 |
| SPPM-AS | 0 | 0 | $\sigma_{\star,\mathrm{AS}}^2$ | 0 | 0 | 0 | Lemma 4 |
| SPPM* | 0 | 0 | 0 | 0 | 0 | 0 | Lemma 5 |
| SPPM-GC | $\delta^2$ | 0 | 0 | 0 | 0 | 0 | Lemma 6 |
| L-SVRP | 0 | $\delta^2$ | 0 | $p$ | $1-p$ | 0 | Lemma 7 |
| Point SAGA | 0 | $\nu^2$ | 0 | $\frac{1}{n}$ | $\frac{n-1}{n}$ | 0 | Lemma 8 |

Table 2: The parameters determining the compliance of the methods listed in Table 1 to Assumption 7. Detailed explanations of the expressions featured in the table are provided in the Appendix.

# 6 EXPERIMENTS

In this section we describe numerical experiments conducted for the linear regression problem

$$\min_{x \in \mathbb{R}^d} \left\{ \frac{1}{2n} \sum_{i=1}^{n} (a_i^\top x - b_i)^2 + \lambda_i \|x\|^2 \right\}, \tag{17}$$

where $a_i \in \mathbb{R}^d$, $b_i \in \mathbb{R}^d$ is the $i$-th data pair, each $\lambda_i$ is a $\ell_2$-regularization parameter. We provide 4 sets of experiments. The first one is devoted to the comparison of different sampling strategies for SPPM-NS : uniform sampling US, importance sampling IS, variance sampling VS (see Section B.3), with three selected stepsizes. The second set of experiments demonstrates the change of the radius of neighborhood for SPPM-AS with $\tau$-nice sampling. In the next two sets of experiments we illustrate the main difference between SPPM and SPPM with variance reduction. Also in practice we show the relationship between SPPM-GC, L-SVRP, Poin-SAGA as we did in theory. For the first bunch of experiments we set $n = 10$, $d = 3$ and the regularization parameters $\lambda_i = 1/2^i$, where $i \in [d]$. Looking at Figure 1, we can see the different behavior of the methods. In the first two plots with the smallest stepsizes, SPPM-IS has a faster start, but a larger neighborhood than the other considered methods. Unfortunately, we cannot say that SPPM-VS has a much smaller neighborhood radius than SPPM-US or SPPM-IS, but theoretically, the variance is smaller. In the second set of numerical experiments (see Figure 2), we observe a clear correlation between the neighborhood radius and the cardinality of the sampled subset $\tau$. More precisely, the larger $\tau$ is, the smaller the neighborhood radius is. In the third set of numerical experiments (see Figure 3), we set $n = 1000$, $d = 10$ and each $\lambda_i = 1$ for the problem (17). On all three plots we observe the superiority of SPPM-star in terms of the stepsize choice. For example, in the third plot with $\gamma = 10^2$, SPPM-GC diverges. In the fourth set of numerical experiments (see Figure 4) with parameters $n = 1000$, $d = 10$ and each $\lambda_i = 1$ for the problem (17), we observe that the performances of SPPM-GC and L-SVRP with $p = 1$ are identical, which is supported by our theoretical findings. With decreasing $p$ from 1 to $1/n$ we see how the behavior of L-SVRP worsens and matches with the performance of Point-SAGA when $p = 1/n$.

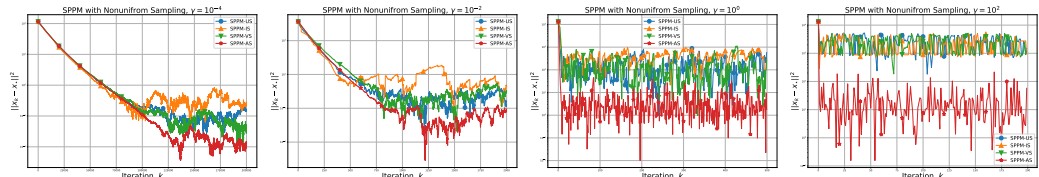

Figure 1: Comparison of the performance of SPPM-US, SPPM-IS, SPPM-VS and SPPM-AS with $\tau = 9$-nice sampling for different selections of stepsize $\gamma \in \{10^{-4}, 10^{-2}, 1, 10^2\}$.

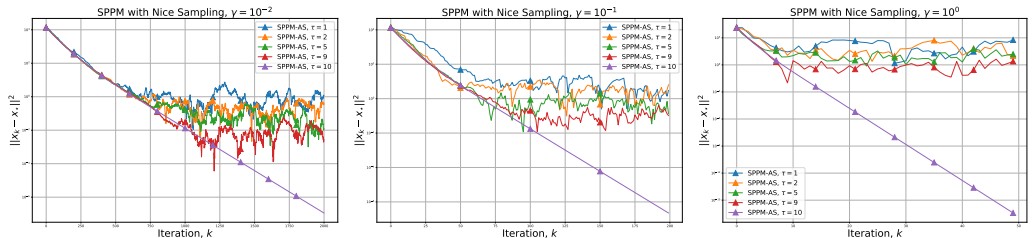

Figure 2: Comparison of the performance of SPPM-AS with $\tau$-nice sampling with different selections of cardinality $\tau \in \{1, 2, 5, 9, n = 10\}$ and stepsize $\gamma \in \{10^{-2}, 10^{-1}, 1\}$.

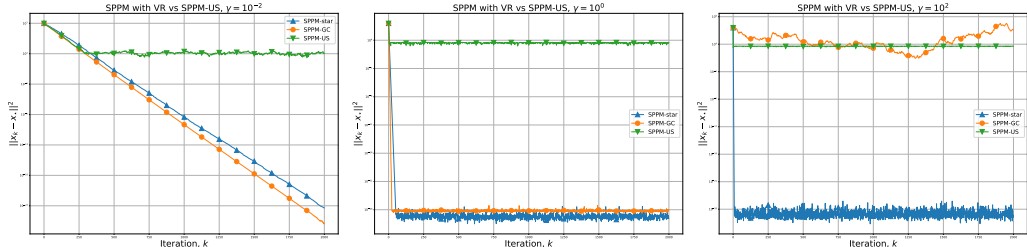

Figure 3: Comparison of the performance of SPPM-US , SPPM-GC and SPPM-star with different selections of stepsize $\gamma \in \{10^{-2}, 1, 10^2\}$.

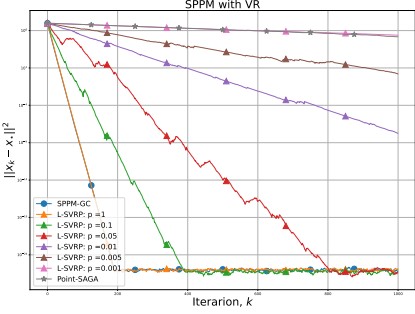

Figure 4: Comparison of the performance of SPPM-GC , Point-SAGA , L-SVRP with different selection of probabilities $\gamma \in \{1/n = 10^{-3}, 5 \cdot 10^{-3}, 10^2, 5 \cdot 10^2, 10^{-1}, 1\}$ . The stepsizes are taken according to the theory.

## 7 FURTHER DISCUSSION

Although our approach is general, we still see some limitations, open problems and several possible directions for future extensions. Generating a similar result in the nonconvex case continues to be an unsolved challenge. Expanding Assumption 7 to incorporate iteration-dependent parameters $A_1, B_1, C_1, A_2, B_2, C_2$ could facilitate the development of various novel methods as SPPM with

decreasing stepsizes. Khaled and Jin (2023) explore federated learning, investigating client sampling to enhance communication efficiency. Another way to do that is to incorporate the utilization of compressed vectors exchanged between a server and the clients. This motivates the problem of providing the analysis for such SPPM-type methods and incorporating them into our framework. It would be interesting to build theory for algorithms with correction vectors $h_k$ with a non-zero expected value and unify with our theory. Another potential avenue for future research involves offering a comprehensive analysis of SPPM-type methods incorporating acceleration and momentum.

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

CONTENTS

## A    EXTENDED LITERATURE OVERVIEW

Stochastic gradient descent (SGD) Robbins and Monro (1951); Nemirovski et al. (2009); Bottou (2010) is a contemporary common algorithm for solving optimization problems (1) and (3) with updates of the form $x^{k+1} = x^k - \gamma g^k$, where $g^k$ is an unbiased stochastic gradient estimator: $\mathrm{E}\left[g^k | x^k\right] = \nabla f(x^k)$. Theoretical properties of SGD are nowadays studied well in works by Vaswani et al. (2019); Gower et al. (2019); Gorbunov et al. (2020); Khaled and Richtárik (2023); Demidovich et al. (2023). The versatility in the design of the estimator $g^k$ has led to a development of highly effective variants of SGD based on importance sampling (see works by Needell et al. (2015); Zhao and Zhang (2015)), mini-batching (see a work of Konečný et al. (2016)). These strategies are unified in the arbitrary sampling paradigm proposed by Gower et al. (2019). The iterates of standard SGD as well as of its sampling variants do not converge to the optimum due to the presence of the stochastic gradient noise. The existence of this issue led to the construction of variance reduction methods in works by Roux et al. (2012); Defazio et al. (2014); Johnson and Zhang (2013); Kovalev et al. (2020); Nguyen et al. (2017). Such methods are able to sequentially learn the stochastic gradients at the optimum which allows them to reach a linear convergence to $x_\star$ when equipped with constant stepsize in case of strongly convex $f$. The work by Gorbunov et al. (2020) provided a unified theory for SGD and its variance-reduced, sampling, quantization and coordinate sub-sampling modifications. Problem setting (3) can also be used to describe a Federated Learning setup where $n$ is the number of workers, and $f_i$ stands for the loss dependent on the data of worker $i \in [n]$. Workers compute individual stochastic gradients and aggregate them on a master node (see a paper by Konečný and Richtárik (2018)). A primary bottleneck in such settings is communication. To address it, many techniques are used as quantization (see works by Gupta et al. (2015); Seide et al. (2014)), sparsification (see a paper by Alistarh et al. (2018)), dithering (see a work by Alistarh et al. (2017)). Many distributed optimization works employ variants of Assumption 5 to analyze methods that solve the problem (3) in the strongly convex case (see works by Shamir et al. (2014); Zhang and Xiao (2015); Yuan and Li (2020); Karimireddy et al. (2020)). Szlendak et al. (2022); Beznosikov and Gasnikov (2022); Panferov et al. (2024) achieve better communication complexity guarantees in a Federated Learning setting under the similarity assumption employing specific quantizers and sparsifiers. Selecting the right stepsize is a critical aspect in the implementation of SGD, see the work of Bach and Moulines (2011). In Section 2 we show that the iteration complexity of SGD depends on condition number $\kappa = \frac{L}{\mu}$. If we are able to evaluate stochastic proximal operator, another algorithm to use instead of SGD is stochastic proximal point method (SPPM). Its iteration complexity does not depend on the smoothness constant $L$. Bertsekas (2011) calls it the incremental proximal point method and considers it for solving the problem (3), proving a convergence to the neighborhood when each $f_i$ is Lipschitz. In the study by Ryu and Boyd (2016), convergence rates are presented for SPPM, emphasizing its robustness against learning rate misspecification, a characteristic not exhibited by SGD. Pătrașcu and Necoara (2017) consider SPPM to solve a general stochastic optimization problem (1) in the convex case and provide nonasymptotic convergence guarantees. Asi and Duchi (2019) delve into the exploration of a broader method termed AProx, which includes SPPM as a particular case. Their research entails analysis of convergence rates for convex functions. Variance-reduced versions of SPPM should also have advantages over their SGD counterparts. It motivated Khaled and Jin (2023) to analyze L-SVRP in the finite-sum and federated settings under a stronger version of Assumption 5 and strong convexity. Defazio (2016) analyzes Point SAGA under the individual smoothness and strong convexity assumptions. Traoré et al. (2023) consider variance-reduced methods L-SVRP and Point SAGA when each $f_i(x)$ is $L$-smooth, convex, and either $f(x)$ is convex or satisfies PŁ-condition. Kim et al. (2022) consider SPPM with momentum.

## B    SPECIAL CASES

### B.1    STOCHASTIC PROXIMAL POINT METHOD WITH LEARNED CORRECTION (SPPM-LC)

This section is devoted to a unified analysis of all stochastic proximal point methods we have encountered so far: SPPM, SPPM-NS, SPPM-AS, SPPM*, SPPM-GC, L-SVRP and Point-SAGA. The analysis is based on formulating a new parametric assumption (Assumption 7) on the behavior of the method, and the convergence result then follows. All that has to be done is to check that this parametric assumption holds in each particular case of interest.

**Lemma 1.** *Assumption 7 holds with constants $A_1, B_1, C_1$ and $A_2, B_2, C_2$.*

*Proof of Lemma 1.* Since we assume the iterates produced by SPPM-LC (Algorithm 1) satisfy Assumption 7, the statement of Lemma 1 holds automatically. $\square$

We can now present the main result of this section.

**Theorem 1.** *Let Assumption 1 (differentiability) and Assumption 2 ($\mu$-strong convexity) hold. Let $\{x_k, h_k\}$ be the iterates produced by SPPM-LC (Algorithm 1), and assume that they satisfy Assumption 7 ($\sigma_k^2$). Choose any $\gamma > 0$ and $\alpha > 0$ satisfying the inequalities*

$$\frac{(1+\gamma^2 A_1)(1+\alpha A_2)}{(1+\gamma\mu)^2} < 1, \qquad \frac{\gamma^2 B_1(1+\alpha A_2)}{\alpha(1+\gamma\mu)^2} + B_2 < 1, \tag{18}$$

*and define the Lyapunov function*

$$\Psi_k := \|x_k - x_\star\|^2 + \alpha\sigma_k^2. \tag{19}$$

*Then for all iterates $k \geq 0$ of SPPM-LC we have*

$$\mathrm{E}\left[\Psi_k\right] \leq \theta^k \Psi_0 + \frac{\zeta}{1-\theta},$$

*where the parameters $0 \leq \theta < 1$ and $\zeta \geq 0$ are given by*

$$\theta = \max\left\{\frac{(1+\gamma^2 A_1)(1+\alpha A_2)}{(1+\gamma\mu)^2}, \frac{\gamma^2 B_1(1+\alpha A_2)}{\alpha(1+\gamma\mu)^2} + B_2\right\}, \qquad \zeta = \frac{\gamma^2 C_1(1+\alpha A_2)}{(1+\gamma\mu)^2} + \alpha C_2.$$

*Proof of Theorem 1.* We use Lemma 9 (see Appendix C) and $\sigma_k^2$-assumption. In particular, by combining inequality Assumption 7 (see (9)) and Lemma 9 (see (70)), for all $\gamma > 0$ we get

$$
\begin{aligned}
\mathrm{E}\left[\Psi_{k+1}\right] &\overset{(19)}{=} \mathrm{E}\left[\|x_{k+1} - x_\star\|^2 + \alpha\sigma_{k+1}^2\right] \\
&= \mathrm{E}\left[\|x_{k+1} - x_\star\|^2\right] + \alpha\mathrm{E}\left[\sigma_{k+1}^2\right] \\
&\overset{(70)}{\leq} \frac{(1+\gamma^2 A_1)}{(1+\gamma\mu)^2}\mathrm{E}\left[\|x_k - x_\star\|^2\right] + \frac{\gamma^2 B_1}{(1+\gamma\mu)^2}\mathrm{E}\left[\sigma_k^2\right] + \frac{\gamma^2 C_1}{(1+\gamma\mu)^2} + \alpha\mathrm{E}\left[\sigma_{k+1}^2\right] \\
&\overset{(8)}{\leq} \frac{(1+\gamma^2 A_1)}{(1+\gamma\mu)^2}\mathrm{E}\left[\|x_k - x_\star\|^2\right] + \frac{\gamma^2 B_1}{(1+\gamma\mu)^2}\mathrm{E}\left[\sigma_k^2\right] + \frac{\gamma^2 C_1}{(1+\gamma\mu)^2} \\
&\qquad + \alpha A_2\mathrm{E}\left[\|x_{k+1} - x_\star\|^2\right] + \alpha B_2\mathrm{E}\left[\sigma_k^2\right] + \alpha C_2 \\
&\overset{(70)}{\leq} \frac{(1+\gamma^2 A_1)}{(1+\gamma\mu)^2}\mathrm{E}\left[\|x_k - x_\star\|^2\right] + \frac{\gamma^2 B_1}{(1+\gamma\mu)^2}\mathrm{E}\left[\sigma_k^2\right] + \frac{\gamma^2 C_1}{(1+\gamma\mu)^2} \\
&\qquad + \frac{(1+\gamma^2 A_1)\alpha A_2}{(1+\gamma\mu)^2}\mathrm{E}\left[\|x_k - x_\star\|^2\right] + \frac{\gamma^2 B_1\alpha A_2}{(1+\gamma\mu)^2}\mathrm{E}\left[\sigma_k^2\right] + \frac{\gamma^2 C_1\alpha A_2}{(1+\gamma\mu)^2} \\
&\qquad + \alpha B_2\mathrm{E}\left[\sigma_k^2\right] + \alpha C_2 \\
&= \frac{(1+\gamma^2 A_1)(1+\alpha A_2)}{(1+\gamma\mu)^2}\mathrm{E}\left[\|x_k - x_\star\|^2\right] + \left(\frac{\gamma^2 B_1(1+\alpha A_2)}{\alpha(1+\gamma\mu)^2} + B_2\right)\alpha\mathrm{E}\left[\sigma_k^2\right] \\
&\qquad + \frac{\gamma^2 C_1(1+\alpha A_2)}{(1+\gamma\mu)^2} + \alpha C_2 \\
&\leq \underbrace{\max\left\{\frac{(1+\gamma^2 A_1)(1+\alpha A_2)}{(1+\gamma\mu)^2}, \frac{\gamma^2 B_1(1+\alpha A_2)}{\alpha(1+\gamma\mu)^2} + B_2\right\}}_{:=\theta}\mathrm{E}\left[V_k\right] \\
&\qquad + \underbrace{\frac{\gamma^2 C_1(1+\alpha A_2)}{(1+\gamma\mu)^2} + \alpha C_2}_{:=\zeta} \\
&= \theta\mathrm{E}\left[\Psi_k\right] + \zeta.
\end{aligned}
$$

By unrolling the recurrence (we can also just apply Fact 3), we get

$$\mathrm{E}\left[\Psi_k\right] \le \theta^k \Psi_0 + \sum_{t=0}^{k-1} \theta^{k-1-t}\zeta = \theta^k \Psi_0 + \sum_{t=0}^{k-1}\theta^t\zeta \le \theta^k \Psi_0 + \sum_{t=0}^{\infty}\theta^t\zeta = \theta^k \Psi_0 + \frac{1}{1-\theta}\zeta.$$

$\square$

### B.2 STOCHASTIC PROXIMAL POINT METHOD (SPPM)

We aim to solve problem (1) via the stochastic proximal point method (SPPM), formalized as Algorithm 2. The main step has the form

$$\mathrm{prox}_{\gamma f_{\xi_k}}(x_k) := \arg\min_{x \in \mathbb{R}^d}\left\{f_{\xi_k}(x) + \frac{1}{2\gamma}\|x - x_k\|^2\right\},$$

where $\gamma > 0$ is a stepsize.

---

**Algorithm 2** Stochastic Proximal Point Method (SPPM)

1: **Parameters:** learning rate $\gamma > 0$, starting point $x_0 \in \mathbb{R}^d$
2: **for** $k = 0, 1, 2, \ldots$ **do**
3:    Sample $\xi_k \sim \mathcal{D}$
4:    $x_{k+1} = \mathrm{prox}_{\gamma f_{\xi_k}}(x_k)$
5: **end for**

---

Notice that if $\gamma$ is kept "too large", then $x_{k+1} \approx \arg\min_x f_{\xi_k}(x)$, and hence the method just "samples" the minimizers of the stochastic functions, and does not progress towards finding $x_\star$ (unless, of course, $x_\star$ happens to be shared by all functions $f_{\xi_k}$). As we shall see, the situation is different if $\gamma$ is kept sufficiently small.

**Lemma 2** (SPPM). *Assumption 7 holds for the iterates of* SPPM *(Algorithm 2) with*

$$A_1 = 0, B_1 = 0, C_1 = \sigma_\star^2, \quad and \quad A_2 = 0, B_2 = 0, C_2 = 0.$$

*Proof of Lemma 2.* Recall that the iterates of SPPM have the form

$$x_{k+1} = \mathrm{prox}_{\gamma f_{\xi_k}}(x_k).$$

Thus, $h_k = 0$. Let $\sigma^2 \equiv 0$. Clearly, (7) holds. Furthermore,

$$\mathrm{E}\left[\|h_k - \nabla f_{\xi_k}(x_\star)\|^2\,\Big|\,x_k, \sigma_k^2\right] = \mathrm{E}\left[\|\nabla f_{\xi_k}(x_\star)\|^2\right] := \sigma_\star^2.$$

Hence, Assumption 7 holds with

$$A_1 = 0, B_1 = 0, C_1 = \sigma_\star^2, \quad and \quad A_2 = 0, B_2 = 0, C_2 = 0.$$

$\square$

The convergence of SPPM is captured by the following theorem.

**Theorem 2.** *Let Assumption 1 (differentiability) and Assumption 2 ($\mu$-strong convexity) hold and define*

$$\sigma_\star^2 := \mathrm{E}_{\xi \sim \mathcal{D}}\left[\|\nabla f_\xi(x_\star)\|^2\right]. \tag{20}$$

*Let $x_0 \in \mathbb{R}^d$ be an arbitrary starting point. Then for any $k \ge 0$ and any $\gamma > 0$, the iterates of* SPPM *(Algorithm 2) satisfy*

$$\mathrm{E}\left[\|x_k - x_\star\|^2\right] \le \left(\frac{1}{1+\gamma\mu}\right)^{2k}\|x_0 - x_\star\|^2 + \frac{\gamma\sigma_\star^2}{\gamma\mu^2 + 2\mu}. \tag{21}$$

Commentary:

1. **Interpolation regime.** Consider the interpolation regime, characterized by $\sigma_\star^2 = 0$. Since we can use arbitrarily large $\gamma > 0$, we obtain an arbitrarily fast convergence rate:

$$\mathrm{E}\left[\|x_k - x_\star\|^2\right] \leq \left(\frac{1}{1+\gamma\mu}\right)^{2k} \|x_0 - x_\star\|^2. \tag{22}$$

Indeed, $\left(\frac{1}{1+\gamma\mu}\right)^{2k}$ can be made arbitrarily small for any fixed $k \geq 1$, even $k = 1$, by choosing $\gamma$ large enough. However, this is not surprising, since now $f$ and all functions $f_\xi$ share a single minimizer, $x_\star$, and hence it is possible to find it by sampling a single function $f_\xi$ and minimizing it, which is what the prox does, as long as $\gamma$ is large enough.

2. **A single step travels far.** Observe that for $\gamma = \frac{1}{\mu}$, we have $\frac{\gamma\sigma_\star^2}{\gamma\mu^2+2\mu} = \frac{\sigma_\star^2}{3\mu^2}$. In fact, the convergence neighborhood $\frac{\gamma\sigma_\star^2}{\gamma\mu^2+2\mu}$ is bounded above by three times this quantity irrespective of the choice of the stepsize. Indeed,

$$\frac{\gamma\sigma_\star^2}{\gamma\mu^2 + 2\mu} \leq \min\left\{\frac{\sigma_\star^2}{\mu^2}, \frac{\gamma\sigma_\star^2}{\mu}\right\} \leq \frac{\sigma_\star^2}{\mu^2}.$$

This means that no matter how far the starting point $x_0$ is from the optimal solution $x_\star$, if we choose the stepsize $\gamma$ to be large enough, then we can get a decent-quality solution after a single iteration of SPPM already! Indeed, if we choose $\gamma$ large enough so that

$$\left(\frac{1}{1+\gamma\mu}\right)^2 \|x_0 - x_\star\|^2 \leq \delta,$$

where $\delta > 0$ is chosen arbitrarily, then for $k = 1$ we get

$$\mathrm{E}\left[\|x_1 - x_\star\|^2\right] \quad \leq \quad \delta + \frac{\sigma_\star^2}{\mu^2}.$$

3. **Iteration complexity.** We have seen above that accuracy arbitrarily close to (but not reaching) $\frac{\sigma_\star^2}{\mu^2}$ can be achieved via a single step of the method, provided the stepsize $\gamma$ is large enough. Assume now that we aim for $\varepsilon$ accuracy where $\varepsilon \leq \frac{\sigma_\star^2}{\mu^2}$. Using the inequality $1 - t \leq \exp(-t)$ which holds for all $t > 0$, we get

$$\left(\frac{1}{1+\gamma\mu}\right)^{2k} = \left(1 - \frac{\gamma\mu}{1+\gamma\mu}\right)^{2k} \leq \exp\left(-\frac{2\gamma\mu k}{1+\gamma\mu}\right).$$

Therefore, provided that

$$k \geq \frac{1+\gamma\mu}{2\gamma\mu} \log\left(\frac{2\|x_0 - x_\star\|^2}{\varepsilon}\right),$$

we get $\left(\frac{1}{1+\gamma\mu}\right)^{2k}\|x_0 - x_\star\|^2 \leq \frac{\varepsilon}{2}$. Furthermore, as long as $\gamma \leq \frac{2\varepsilon\mu}{2\sigma_\star^2 - \varepsilon\mu^2}$ (this is true provided that the more restrictive but also more elegant-looking condition $\gamma \leq \varepsilon\frac{\mu}{\sigma_\star^2}$ holds), we get

$$\frac{\gamma\sigma_\star^2}{\gamma\mu^2 + 2\mu} \leq \frac{\varepsilon}{2}.$$

Putting these observations together, we conclude that with the stepsize $\gamma = \varepsilon\frac{\mu}{\sigma_\star^2}$, we get

$$\mathrm{E}\left[\|x_k - x_\star\|^2\right] \leq \varepsilon$$

provided that

$$k \geq \frac{1+\gamma\mu}{2\gamma\mu} \log\frac{2\|x_0 - x_\star\|^2}{\varepsilon} = \left(\frac{\sigma_\star^2}{2\varepsilon\mu^2} + \frac{1}{2}\right) \log\left(\frac{2\|x_0 - x_\star\|^2}{\varepsilon}\right). \tag{23}$$

*Proof of Theorem 2.* Recall that in the case of SPPM, we have $A_1 = 0, B_1 = 0, C_1 = \sigma_\star^2, A_2 = 0, B_2 = 0, C_2 = 0$ (it is the result of Lemma 2). From Theorem 1, choosing any $\alpha > 0$, $\theta = \frac{1}{(1+\gamma\mu)^2}, \zeta = \frac{\gamma^2\sigma_\star^2}{(1+\gamma\mu)^2}$ (see (15) and (16)), we obtain

$$\frac{\zeta}{1-\theta} = \frac{\gamma^2\sigma_\star^2}{(1+\gamma\mu)^2(1-\theta)} = \frac{\gamma^2\sigma_\star^2}{(1+\gamma\mu)^2-1} = \frac{\gamma\sigma_\star^2}{\gamma\mu^2+2\mu}.$$

For any $k \geq 0$ and any $\gamma > 0$, the iterates of SPPM (Algorithm 2) satisfy

$$\mathrm{E}\left[\|x_k - x_\star\|^2\right] \leq \left(\frac{1}{1+\gamma\mu}\right)^{2k}\|x_0 - x_\star\|^2 + \frac{\gamma\sigma_\star^2}{\gamma\mu^2+2\mu}.$$

$\square$

### B.3 STOCHASTIC PROXIMAL POINT METHOD WITH NONUNIFORM SAMPLING (SPPM-NS)

Applying SPPM directly to the optimization formulation (3), Theorem 2 applies with $\mu = \min_i \mu_i$ since $f$ is $\mu$-strongly convex for $\mu = \min_i \mu_i$. If $\min_i \mu_i$ is small, the convergence rate becomes weak. In this section we shall describe a trick which enables us to obtain a dependence on the average of the strong convexity constants instead of their minimum.

Choose positive numbers $p_1, \ldots, p_n$ summing up to 1, and let

$$\tilde{f}_i(x) := \frac{1}{np_i}f_i(x), \qquad i \in [n].$$

Note that (3) can be reformulated in the form

$$\min_{x \in \mathbb{R}^d}\left\{f(x) := \sum_{i=1}^n p_i\tilde{f}_i(x)\right\}. \tag{24}$$

We rely on Assumptions 3 and 4.

This is a more refined version of Assumption 2, where we assumed the strong convexity parameter was the same for all functions. This implies that $f$ is $\mu$-strongly convex with $\mu = \min_i \mu_i$. Hence, $f$ has a unique minimizer, which we shall denote $x_\star$.

We can now apply SPPM to the reformulated problem (24) instead, with $\mathcal{D}$ being the nonuniform distribution over the finite set $[n]$ given by the parameters $p_1, \ldots, p_n$ as follows: $\xi = i$ with probability $p_i > 0$. This method is called *stochastic proximal point method with nonuniform sampling* (SPPM-NS):

$$x_{k+1} = \mathrm{prox}_{\gamma\tilde{f}_{i_k}}(x_k) = \mathrm{prox}_{\frac{\gamma}{np_i}f_{i_k}}(x_k),$$

where $i_k = i$ with probability $p_i > 0$.

---

**Algorithm 3** Stochastic Proximal Point Method with Nonuniform Sampling (SPPM-NS)

---

1: **Parameters:** learning rate $\gamma > 0$, starting point $x_0 \in \mathbb{R}^d$, positive probabilities $p_1, \ldots, p_n$ summing up to 1
2: **for** $k = 0, 1, 2, \ldots$ **do**
3:     Choose $i_k = i$ with probability $p_i > 0$
4:     $x_{k+1} = \mathrm{prox}_{\frac{\gamma}{np_i}f_{i_k}}(x_k)$
5: **end for**

---

Define

$$\mu_{\mathrm{NS}} := \min_i \frac{\mu_i}{np_i}, \qquad \sigma_{\star,\mathrm{NS}}^2 := \frac{1}{n}\sum_{i=1}^n \frac{1}{np_i}\|\nabla f_i(x_\star)\|^2. \tag{25}$$

**Lemma 3** (SPPM-NS). *Assumption 7 holds for the iterates of SPPM-NS (Algorithm 3) with*

$$A_1 = 0, B_1 = 0, C_1 = \sigma_{\star,\mathrm{NS}}^2, \quad and \quad A_2 = 0, B_2 = 0, C_2 = 0.$$

*Proof.* Recall that the iterates of (SPPM-NS) have the form

$$x_{k+1} = \mathrm{prox}_{\gamma \tilde{f}_{i_k}}(x_k) = \mathrm{prox}_{\frac{\gamma}{np_i}f_{i_k}}(x_k),$$

where $i_k = i$ with probability $p_i > 0$. Therefore, that $h_k = 0$. Let $\sigma^2 \equiv 0$. Clearly, (7) holds. Furthermore,

$$\mathrm{E}\left[\left\|h_k - \nabla \tilde{f}_{i_k}(x_\star)\right\|^2 \,\Big|\, x_k, \sigma_k^2\right] = \mathrm{E}\left[\left\|\nabla \tilde{f}_{i_k}(x_\star)\right\|^2\right] := \sigma_{\star,\mathrm{NS}}^2.$$

Hence, Assumption 7 holds with

$$A_1 = 0, B_1 = 0, C_1 = \sigma_{\star,\mathrm{NS}}^2, \quad \text{and} \quad A_2 = 0, B_2 = 0, C_2 = 0.$$

$\square$

**Theorem 3.** *Let Assumption 3 and Assumption 4 hold. Let $x_0 \in \mathbb{R}^d$ be an arbitrary starting point. Let $\mu_{\mathrm{NS}}$ and $\sigma_{\star,\mathrm{NS}}^2$ be as in (25). Then for any $k \geq 0$ and any $\gamma > 0$, the iterates of* SPPM-NS *(Algorithm 3) satisfy*

$$\mathrm{E}\left[\|x_k - x_\star\|^2\right] \leq \left(\frac{1}{1+\gamma\mu_{\mathrm{NS}}}\right)^{2k}\|x_0 - x_\star\|^2 + \frac{\gamma\sigma_{\star,\mathrm{NS}}^2}{\gamma\mu_{\mathrm{NS}}^2 + 2\mu_{\mathrm{NS}}}. \tag{26}$$

Commentary:

(a) **Uniform sampling.** If we choose $p_i = \frac{1}{n}$ for all $i \in [n]$, we shall refer to Algorithm 3 as *stochastic proximal point method with uniform sampling* (SPPM-US). In this case,

$$\mu_{\mathrm{NS}} = \mu_{\mathrm{US}} := \min_i \mu_i, \qquad \sigma_{\star,\mathrm{NS}}^2 = \sigma_{\star,\mathrm{US}}^2 := \frac{1}{n}\sum_{i=1}^n \|\nabla f_i(x_\star)\|^2.$$

(b) **Importance sampling: optimizing the linear rate.** If we choose $p_i = \frac{\mu_i}{\sum_{j=1}^n \mu_j}$ for all $i \in [n]$, we shall refer to Algorithm 3 as *stochastic proximal point method with importance sampling* (SPPM-IS). In this case,

$$\mu_{\mathrm{NS}} = \mu_{\mathrm{IS}} := \frac{1}{n}\sum_{i=1}^n \mu_i, \qquad \sigma_{\star,\mathrm{NS}}^2 = \sigma_{\star,\mathrm{IS}}^2 := \frac{\sum_{i=1}^n \mu_i}{n}\sum_{i=1}^n \frac{\|\nabla f_i(x_\star)\|^2}{n\mu_i}.$$

This choice maximizes the value of $\mu_{\mathrm{NS}}$ (and hence the first part of the convergence rate) over the choice of the probabilities.

(c) **Variance sampling: optimizing the variance.** If we choose $p_i = \frac{\|\nabla f_i(x_\star)\|}{\sum_{j=1}^n \|\nabla f_j(x_\star)\|}$ for all $i \in [n]$, we shall refer to Algorithm 3 as *stochastic proximal point method with variance sampling* (SPPM-VS). In this case,

$$\mu_{\mathrm{NS}} = \mu_{\mathrm{VS}} := \frac{1}{n}\sum_{i=1}^n \|\nabla f_i(x_\star)\| \left(\min_i \frac{\mu_i}{\|\nabla f_i(x_\star)\|}\right),$$

$$\sigma_{\star,\mathrm{NS}}^2 = \sigma_{\star,\mathrm{VS}}^2 := \left(\frac{1}{n}\sum_{i=1}^n \|\nabla f_i(x_\star)\|\right)^2.$$

This choice minimizes the value of $\sigma_{\star,\mathrm{NS}}$ (and hence the second part of the convergence rate) over the choice of the probabilities.

*Proof of Theorem 3.* From Lemma 3, we have that Assumption 7 holds for the iterates of SPPM-NS (Algorithm 3) with

$$A_1 = 0, B_1 = 0, C_1 = \sigma_{\star,\mathrm{NS}}^2, \quad \text{and} \quad A_2 = 0, B_2 = 0, C_2 = 0.$$

From Theorem 1, choosing any $\alpha > 0$, $\theta = \frac{1}{(1+\gamma\mu_{\mathrm{NS}})^2}$, $\zeta = \frac{\gamma^2\sigma_{\star,\mathrm{NS}}^2}{(1+\gamma\mu_{\mathrm{NS}})^2}$ (see (15) and (16)), we obtain that, for any $k \geq 0$ and any $\gamma > 0$, the iterates of SPPM-NS (Algorithm 3) satisfy

$$\mathrm{E}\left[\|x_k - x_\star\|^2\right] \leq \left(\frac{1}{1+\gamma\mu_{\mathrm{NS}}}\right)^{2k}\|x_0 - x_\star\|^2 + \frac{\gamma\sigma_{\star,\mathrm{NS}}^2}{\gamma\mu_{\mathrm{NS}}^2 + 2\mu_{\mathrm{NS}}}.$$

Alternatively, we can derive Theorem 3 from Theorem 2. It is easy to show that each function $\tilde{f}_i$ is $\tilde{\mu}_i$-strongly convex with $\tilde{\mu}_i := \frac{\mu_i}{np_i}$. Hence, $\tilde{f}_i$ is also $\mu_{\mathrm{NS}}$-strongly convex for all $i$. It now only remains to apply Theorem 2. $\square$

### B.4 STOCHASTIC PROXIMAL POINT METHOD WITH ARBITRARY SAMPLING (SPPM-AS)

We consider the optimization problem (3) and rely on Assumption 3 (differentiability) and Assumption 4 (strong convexity).

Let $\mathcal{S}$ be a probability distribution over the $2^n$ subsets of $[n]$. Given a random set $S \sim \mathcal{S}$, we define

$$p_i := \mathbf{Prob}\left(i \in S\right), \qquad i \in [n]. \tag{27}$$

We will restrict our attention to proper and nonvacuous random sets.

**Assumption 8.** *$S$ is proper (i.e., $p_i > 0$ for all $i \in [n]$) and nonvacuous (i.e., $\mathbf{Prob}\left(S = \varnothing\right) = 0$).*

Given $\varnothing \neq C \subseteq [n]$ and $i \in [n]$, we define

$$v_i(C) := \begin{cases} \frac{1}{p_i} & i \in C \\ 0 & i \notin C \end{cases}, \tag{28}$$

and

$$f_C(x) := \frac{1}{n} \sum_{i=1}^{n} v_i(C) f_i(x) \overset{(28)}{=} \sum_{i \in C} \frac{1}{np_i} f_i(x). \tag{29}$$

Note that $v_i(S)$ is a random variable and $f_S$ is a random function. By construction, $\mathrm{E}_{S \sim \mathcal{S}}\left[v_i(S)\right] = 1$ for all $i \in [n]$, and hence

$$\mathrm{E}_{S \sim \mathcal{S}}\left[f_S(x)\right] = \mathrm{E}_{S \sim \mathcal{S}}\left[\frac{1}{n} \sum_{i=1}^{n} v_i(S) f_i(x)\right] = \frac{1}{n} \sum_{i=1}^{n} \mathrm{E}_{S \sim \mathcal{S}}\left[v_i(S)\right] f_i(x) = \frac{1}{n} \sum_{i=1}^{n} f_i(x) = f(x).$$

Therefore, the optimization problem (3) is equivalent to the stochastic optimization problem

$$\min_{x \in \mathbb{R}^d} \left\{f(x) := \mathrm{E}_{S \sim \mathcal{S}}\left[f_S(x)\right]\right\}. \tag{30}$$

Further, if for each $C \subset [n]$ we let $p_C := \mathbf{Prob}\left(S = C\right)$, $f$ can be written in the equivalent form

$$f(x) = \mathrm{E}_{S \sim \mathcal{S}}\left[f_S(x)\right] = \sum_{C \subseteq [n]} p_C f_C(x) = \sum_{C \subseteq [n], p_C > 0} p_C f_C(x). \tag{31}$$

Applying SPPM to (30), we arrive at *stochastic proximal point method with arbitrary sampling (SPPM-AS)* (Algorithm 4):

$$x_{k+1} = \mathrm{prox}_{\gamma f_{S_k}}\left(x_k\right),$$

where $S_k \sim \mathcal{S}$.

---

**Algorithm 4** Stochastic Proximal Point Method with Arbitrary Sampling (SPPM-AS)

1: **Parameters:** learning rate $\gamma > 0$, starting point $x_0 \in \mathbb{R}^d$, distribution $\mathcal{S}$ over the subsets of $[n]$
2: **for** $k = 0, 1, 2, \ldots$ **do**
3:     Sample $S_k \sim \mathcal{S}$
4:     $x_{k+1} = \mathrm{prox}_{\gamma f_{S_k}}\left(x_k\right)$
5: **end for**

---

Define

$$\mu_{\mathrm{AS}} := \min_{C \subseteq [n], p_C > 0} \sum_{i \in C} \frac{\mu_i}{np_i}, \qquad \sigma_{\star, \mathrm{AS}}^2 := \sum_{C \subseteq [n], p_C > 0} p_C \left\| \sum_{i \in C} \frac{1}{np_i} \nabla f_i(x_\star) \right\|^2. \tag{32}$$

**Lemma 4.** *Assumption 7 holds for the iterates of* SPPM-AS *(Algorithm 4) with*

$$A_1 = 0, B_1 = 0, C_1 = \sigma_{\star, \mathrm{AS}}^2, \quad and \quad A_2 = 0, B_2 = 0, C_2 = 0.$$

*Proof.* Recall that the iterates of SPPM-AS (Algorithm 4) have the form

$$x_{k+1} = \text{prox}_{\gamma f_{S_k}}(x_k).$$

Thus, $h_k = 0$. Let $\sigma^2 \equiv 0$. Clearly, (7) holds. Furthermore,

$$\text{E}\left[\left.\|h_k - \nabla f_C(x_\star)\|^2\right| x_k, \sigma_k^2\right] = \text{E}\left[\|\nabla f_C(x_\star)\|^2\right] := \sigma_{\star,\text{AS}}^2,$$

since

$$\begin{aligned}
\sigma_\star^2 &:= \text{E}_{\xi\sim\mathcal{D}}\left[\|\nabla f_\xi(x_\star)\|^2\right] \\
&\overset{(31)}{=} \sum_{C\subseteq[n],p_C>0} p_C\|\nabla f_C(x_\star)\|^2 \\
&\overset{(29)}{=} \sum_{C\subseteq[n],p_C>0} p_C\left\|\sum_{i\in C}\frac{1}{np_i}\nabla f_i(x_\star)\right\|^2 \\
&:= \sigma_{\star,\text{AS}}^2.
\end{aligned}$$

Hence, Assumption 7 holds with

$$A_1 = 0, B_1 = 0, C_1 = \sigma_{\star,\text{AS}}^2, \quad \text{and} \quad A_2 = 0, B_2 = 0, C_2 = 0.$$

$\square$

The convergence of SPPM-AS is captured by the following theorem.

**Theorem 4.** *Let Assumption 3 (differentiability) and Assumption 4 (strong convexity) hold. Let $S$ be a random set satisfying Assumption 8. Let $x_0 \in \mathbb{R}^d$ be an arbitrary starting point, $\mu_{\text{AS}}$ and $\sigma_{\star,\text{AS}}^2$ be as in (32). Then for any $k \geq 0$ and any $\gamma > 0$, the iterates of SPPM-AS (Algorithm 4) satisfy*

$$\text{E}\left[\|x_k - x_\star\|^2\right] \leq \left(\frac{1}{1+\gamma\mu_{\text{AS}}}\right)^{2k}\|x_0 - x_\star\|^2 + \frac{\gamma\sigma_{\star,\text{AS}}^2}{\gamma\mu_{\text{AS}}^2 + 2\mu_{\text{AS}}}. \tag{33}$$

Commentary:

(a) **Full sampling.** Let $S = [n]$ with probability 1 ("full sampling"; abbreviated as "FS"). Then SPPM-AS applied to (30) becomes PPM for minimizing $f$. So, besides Theorem 2, Theorem 4 is the third theorem capturing the convergence of PPM in the differentiable and strongly convex regime.

Moreover, we have $p_i = 1$ for all $i \in [n]$ and the expressions (32) take on the form

$$\mu_{\text{AS}} = \mu_{\text{FS}} := \frac{1}{n}\sum_{i=1}^n \mu_i, \qquad \sigma_{\star,\text{AS}}^2 = \sigma_{\star,\text{FS}}^2 := 0.$$

Note that $\mu_{\text{FS}}$ a lower bound on the the strong convexity constant of $f$ – one that can be computed from the strong convexity constants $\mu_i$ of the constituent functions $f_i$.

(b) **Nonuniform sampling.** Let $S = \{i\}$ with probability $q_i > 0$, where $\sum_i q_i = 1$. This leads to SPPM-NS. Then $p_i := \textbf{Prob}(i \in S) = q_i$ for all $i \in [n]$, and the expressions (32) take on the form

$$\mu_{\text{AS}} = \mu_{\text{NS}} := \min_i \frac{\mu_i}{np_i}, \qquad \sigma_{\star,\text{AS}}^2 = \sigma_{\star,\text{NS}}^2 := \frac{1}{n}\sum_{i=1}^n \frac{1}{np_i}\|\nabla f_i(x_\star)\|^2,$$

which recovers the quantities from Section B.3; see (25).

(c) **Nice sampling.** Choose $\tau \in [n]$ and let $S$ be a random subset of $[n]$ of size $\tau$ chosen uniformly at random. We call the resulting method SPPM-NICE. Then $p_i := \textbf{Prob}(i \in S) = \frac{\tau}{n}$ for all $i \in [n]$. Moreover, $p_C = \frac{1}{\binom{n}{\tau}}$ whenever $|C| = \tau$ and $p_C = 0$ otherwise. So,

$$\mu_{\text{AS}} = \mu_{\text{NICE}}(\tau) := \min_{C\subseteq[n],|C|=\tau}\frac{1}{\tau}\sum_{i\in C}\mu_i \tag{34}$$

and

$$\sigma^2_{\star,\mathrm{AS}} = \sigma^2_{\star,\mathrm{NICE}}(\tau) := \sum_{C \subseteq [n], |C| = \tau} \frac{1}{\binom{n}{\tau}} \left\| \frac{1}{\tau} \sum_{i \in C} \nabla f_i(x_\star) \right\|^2. \tag{35}$$

It can be shown that $\mu_{\mathrm{NICE}}(\tau)$ is a nondecreasing function of $\tau$. So, as the minibatch size $\tau$ increases, the strong convexity constant $\mu_{\mathrm{NICE}}(\tau)$ can only improve. Since $\mu_{\mathrm{NICE}}(1) = \min_i \mu_i$ and $\mu_{\mathrm{NICE}}(n) = \frac{1}{n} \sum_{i=1}^{n} \mu_i$, the value of $\mu_{\mathrm{NICE}}(\tau)$ interpolates these two extreme cases as $\tau$ varies between $1$ and $n$.

(d) **Block sampling.** Let $C_1, \ldots, C_b$ be a partition of $[n]$ into $b$ nonempty blocks. For each $i \in [n]$, let $B(i)$ indicate which block does $i$ belong to. In other words, $i \in C_j$ iff $B(i) = j$. Let $S = C_j$ with probability $q_j > 0$, where $\sum_j q_j = 1$. We call the resulting method SPPM-BS. Then $p_i := \mathbf{Prob}\,(i \in S) = q_{B(i)}$, and hence the expressions (32) take on the form

$$\mu_{\mathrm{AS}} = \mu_{\mathrm{BS}} := \min_{j \in [b]} \frac{1}{nq_j} \sum_{i \in C_j} \mu_i, \qquad \sigma^2_{\star,\mathrm{AS}} = \sigma^2_{\star,\mathrm{BS}} := \sum_{j \in [b]} q_j \left\| \sum_{i \in C_j} \frac{1}{np_j} \nabla f_i(x_\star) \right\|^2.$$

We now consider two extreme cases:

- If $b = 1$, then SPPM-BS = SPPM-FS = PPM. Let's see, as a sanity check, whether we recover the right rate as well. We have $q_1 = 1$, $C_1 = [n]$, $p_i := \mathbf{Prob}\,(i \in S) = 1$ for all $i \in [n]$, and the expressions for $\mu_{\mathrm{AS}}$ and $\sigma^2_{\star,\mathrm{BS}}$ simplify to

$$\mu_{\mathrm{BS}} = \mu_{\mathrm{FS}} := \frac{1}{n} \sum_{i=1}^{n} \mu_i, \qquad \sigma^2_{\star,\mathrm{BS}} = \sigma^2_{\star,\mathrm{FS}} := 0.$$

  So, indeed, we recover the same rate as SPPM-FS.

- If $b = n$, then SPPM-BS = SPPM-NS. Let's see, as a sanity check, whether we recover the right rate as well. We have $C_i = \{i\}$ and $p_i := \mathbf{Prob}\,(i \in S) = q_i$ for all $i \in [n]$, and the expressions for $\mu_{\mathrm{AS}}$ and $\sigma^2_{\star,\mathrm{BS}}$ simplify to

$$\mu_{\mathrm{BS}} = \mu_{\mathrm{NS}} := \min_{i \in [n]} \frac{\mu_i}{np_i}, \qquad \sigma^2_{\star,\mathrm{BS}} = \sigma^2_{\star,\mathrm{NS}} := \frac{1}{n} \sum_{i=1}^{n} \frac{1}{np_i} \|\nabla f_i(x_\star)\|^2.$$

  So, indeed, we recover the same rate as SPPM-NS.

(e) **Stratified sampling.** Let $C_1, \ldots, C_b$ be a partition of $[n]$ into $b$ nonempty blocks, as before. For each $i \in [n]$, let $B(i)$ indicate which block $i$ belongs to. In other words, $i \in C_j$ iff $B(i) = j$. Now, for each $j \in [b]$ pick $\xi_j \in C_j$ uniformly at random, and define $S = \cup_{j \in [b]} \{\xi_j\}$. We call the resulting method SPPM-SS. Clearly, $p_i := \mathbf{Prob}\,(i \in S) = \frac{1}{|C_{B(i)}|}$. The expressions (32) take on the form

$$\mu_{\mathrm{AS}} = \mu_{\mathrm{SS}} := \min_{(i_1, \ldots, i_b) \in C_1 \times \cdots \times C_b} \sum_{j=1}^{b} \frac{\mu_{i_j} |C_j|}{n}$$

and

$$\sigma^2_{\star,\mathrm{AS}} = \sigma^2_{\star,\mathrm{SS}} := \left( \frac{1}{\prod_{j=1}^{b} |C_j|} \right) \sum_{(i_1, \ldots, i_b) \in C_1 \times \cdots \times C_b} \left\| \sum_{j=1}^{b} \frac{|C_j|}{n} \nabla f_{i_j}(x_\star) \right\|^2.$$

We now consider two extreme cases:

- If $b = 1$, then SPPM-SS = SPPM-US. Let's see, as a sanity check, whether we recover the right rate as well. We have $C_1 = [n]$, $|C_1| = n$, $\left( \prod_{j=1}^{b} \frac{1}{|C_j|} \right) = \frac{1}{n}$ and hence

$$\mu_{\mathrm{SS}} = \mu_{\mathrm{US}} := \min_i \mu_i, \qquad \sigma^2_{\star,\mathrm{SS}} = \sigma^2_{\star,\mathrm{US}} := \frac{1}{n} \sum_{i=1}^{n} \|\nabla f_i(x_\star)\|^2.$$

  So, indeed, we recover the same rate as SPPM-US.

- If $b = n$, then SPPM-SS = SPPM-FS. Let's see, as a sanity check, whether we recover the right rate as well. We have $C_i = \{i\}$ for all $i \in [n]$, $\left(\prod_{j=1}^{b} \frac{1}{|C_j|}\right) = 1$, and hence

$$\mu_{\mathrm{SS}} = \mu_{\mathrm{FS}} := \frac{1}{n}\sum_{i=1}^{n}\mu_i, \qquad \sigma^2_{\star,\mathrm{SS}} = \sigma^2_{\star,\mathrm{FS}} := 0.$$

So, indeed, we recover the same rate as SPPM-FS.

*Proof of Theorem 4.* From Lemma 4 we have that Assumption 7 holds for the iterates of SPPM-AS (Algorithm 4) with

$$A_1 = 0, B_1 = 0, C_1 = \sigma^2_{\star,\mathrm{AS}}, \quad \text{and} \quad A_2 = 0, B_2 = 0, C_2 = 0.$$

From Theorem 1, choosing $\alpha > 0$, $\theta = \frac{1}{(1+\gamma\mu)^2}$, $\zeta = \frac{\gamma^2\sigma^2_{\star,\mathrm{AS}}}{(1+\gamma\mu)^2}$ (see (15) and (16)), we obtain that, for any $k \geq 0$ and any $\gamma > 0$, the iterates of SPPM-AS (Algorithm 4) satisfy

$$\mathrm{E}\left[\|x_k - x_\star\|^2\right] \leq \left(\frac{1}{1+\gamma\mu_{\mathrm{AS}}}\right)^{2k}\|x_0 - x_\star\|^2 + \frac{\gamma\sigma^2_{\star,\mathrm{AS}}}{\gamma\mu^2_{\mathrm{AS}} + 2\mu_{\mathrm{AS}}}.$$

Alternatively, we can derive Theorem 4 from Theorem 2. Let $C$ be any (necessarily nonempty) subset of $[n]$ such that $p_C > 0$. Recall that in view of (29) we have

$$f_C(x) = \sum_{i \in C}\frac{1}{np_i}f_i(x);$$

i.e., $f_C$ is a conic combination of the functions $\{f_i \ : \ i \in C\}$ with weights $w_i = \frac{1}{np_i}$. Since each $f_i$ is $\mu_i$-strongly convex, Lemma 10 says that $f_C$ $\mu_C$-strongly convex with

$$\mu_C := \sum_{i \in C}\frac{\mu_i}{np_i}.$$

So, every such $f_C$ is $\mu$-strongly convex with

$$\mu = \mu_{\mathrm{AS}} := \min_{C \subseteq [n], p_C > 0}\sum_{i \in C}\frac{\mu_i}{np_i}.$$

Further, the quantity $\sigma^2_\star$ from (20) is equal to

$$\begin{aligned}
\sigma^2_\star &:= \mathrm{E}_{\xi\sim\mathcal{D}}\left[\|\nabla f_\xi(x_\star)\|^2\right] \\
&\overset{(31)}{=} \sum_{C \subseteq [n], p_C > 0}p_C\|\nabla f_C(x_\star)\|^2 \\
&\overset{(29)}{=} \sum_{C \subseteq [n], p_C > 0}p_C\left\|\sum_{i \in C}\frac{1}{np_i}\nabla f_i(x_\star)\right\|^2 \\
&:= \sigma^2_{\star,\mathrm{AS}}.
\end{aligned}$$

It now only remains to apply Theorem 2. $\qquad\square$

### B.5 Stochastic proximal point method with optimal gradient correction (SPPM*)

We showed that SPPM converges up to a neighborhood of size $\frac{\gamma\sigma^2_\star}{\gamma\mu^2 + 2\mu}$, where

$$\sigma^2_\star := \mathrm{E}_{\xi\sim\mathcal{D}}\left[\|\nabla f_\xi(x_\star)\|^2\right].$$

We now describe a simple trick which gets rid of the neighborhood when $\sigma^2_\star > 0$. The trick is of a conceptual nature: as is, it is practically useless. However, it will serve as an inspiration for a trick that can be implemented.

---

**Algorithm 5** Stochastic Proximal Point Method with Optimal Gradient Correction (SPPM*)

---

1: **Parameters:** learning rate $\gamma > 0$, starting point $x_0 \in \mathbb{R}^d$
2: **for** $k = 0, 1, 2, \ldots$ **do**
3:    Sample $\xi_k \sim \mathcal{D}$
4:    $h_k = \nabla f_{\xi_k}(x_\star)$
5:    $x_{k+1} = \mathrm{prox}_{\gamma f_{\xi_k}}(x_k + \gamma h_k)$                                     $= \mathrm{prox}_{\gamma \tilde{f}_{\xi_k}}(x_k)$
6: **end for**

---

Let us reformulate the problem by adding a smart zero.

For each $\xi \sim \mathcal{D}$, define
$$\tilde{f}_\xi(x) := f_\xi(x) - \langle \nabla f_\xi(x_\star), x \rangle, \tag{36}$$
and instead of solving (1), consider solving the problem
$$\min_{x \in \mathbb{R}^d} \left\{ \tilde{f}(x) := \mathrm{E}_{\xi \sim \mathcal{D}} \left[ \tilde{f}_\xi(x) \right] \right\}. \tag{37}$$

Observations:

- We do not know $\nabla f_\xi(x_\star)$, and hence formulation (37) is not of practical interest.
- It is easy to see that $f = \tilde{f}$, and hence problems (1) and (37) are equivalent. Indeed,

$$
\begin{aligned}
\tilde{f}(x) \quad &\overset{(37)}{=} \quad \mathrm{E}_{\xi \sim \mathcal{D}} \left[ \tilde{f}_\xi(x) \right] \quad \overset{(36)}{=} \quad \mathrm{E}_{\xi \sim \mathcal{D}} \left[ f_\xi(x) - \langle \nabla f_\xi(x_\star), x \rangle \right] \\
&= \quad \mathrm{E}_{\xi \sim \mathcal{D}} \left[ f_\xi(x) \right] - \mathrm{E}_{\xi \sim \mathcal{D}} \left[ \langle \nabla f_\xi(x_\star), x \rangle \right] \\
&= \quad \mathrm{E}_{\xi \sim \mathcal{D}} \left[ f_\xi(x) \right] - \left\langle \underbrace{\mathrm{E}_{\xi \sim \mathcal{D}} \left[ \nabla f_\xi(x_\star) \right]}_{=\nabla f(x_\star)=0}, x \right\rangle \quad \overset{(1)}{=} \quad f(x).
\end{aligned}
$$

- All stochastic gradients of $\tilde{f}$ at $x_\star$ are zero. Indeed,
$$\nabla \tilde{f}(x) \overset{(36)}{=} \nabla f_\xi(x) - \nabla f_\xi(x_\star),$$
and hence $\nabla \tilde{f}_\xi(x_\star) = 0$.
- It is easy to see that since $f_\xi$ is differentiable and $\mu$-strongly convex, then so is $\tilde{f}_\xi$.

**Hiding the prox.** Further, recall that if for some $x \in \mathbb{R}^d$ and differentiable and convex $\phi$ we let $x_+ := \mathrm{prox}_\phi(x)$, then $x_+ = x - \nabla \phi(x_+)$. Therefore, steps 4 and 5 of the method can be written in the equivalent form
$$x_{k+1} = x_k + \gamma h_k - \gamma \nabla f_{\xi_k}(x_{k+1}) = x_k - \gamma \left( \nabla f_{\xi_k}(x_{k+1}) - \nabla f_{\xi_k}(x_\star) \right).$$

**Lemma 5** (SPPM*)**.** *Assumption 7 holds for the iterates of* SPPM* *(Algorithm 5) with*
$$A_1 = 0, B_1 = 0, C_1 = 0, \quad and \quad A_2 = 0, B_2 = 0, C_2 = 0.$$

*Proof.* Recall that the iterates of SPPM* have the form
$$x_{k+1} = \mathrm{prox}_{\gamma f_{\xi_k}}(x_k + \gamma \nabla f_{\xi_k}(x_\star)).$$

Thus, $h_k = \nabla f_{\xi_k}(x_\star)$. Let $\sigma^2 \equiv 0$. Clearly, (7) holds. Furthermore,
$$\mathrm{E}\left[ \|h_k - \nabla f_{\xi_k}(x_\star)\|^2 \,\Big|\, x_k, \sigma_k^2 \right] = 0.$$

Hence, Assumption 7 holds with
$$A_1 = 0, B_1 = 0, C_1 = 0, \quad and \quad A_2 = 0, B_2 = 0, C_2 = 0.$$

$\square$

The convergence of SPPM* is captured by the following theorem.

**Theorem 5.** *Let Assumption 1 (differentiability), Assumption 2 ($\mu$-strong convexity) hold. Let $x_0 \in \mathbb{R}^d$ be an arbitrary starting point. Then for any $k \geq 0$ and any $\gamma > 0$, the iterates of SPPM\* (Algorithm 5) satisfy*

$$\mathrm{E}\left[\|x_k - x_\star\|^2\right] \leq \left(\frac{1}{1 + \gamma\mu}\right)^{2k} \|x_0 - x_\star\|^2. \tag{38}$$

Commentary:

- The convergence neighborhood is fully removed; the method converges to the exact solution! The result is identical to (22); i.e., to the rate of SPPM in the interpolation regime.
    - The method converges to $x_\star$ for any fixed $\gamma > 0$ as long as $k \to \infty$.
    - The method converges to $x_\star$ for any fixed $k \geq 1$ as long as $\gamma \to \infty$.
- The method is practically useless since it relies on the knowledge of the optimal stochastic gradients $\nabla f_\xi(x_\star)$ for all $\xi$ as hyper-parameters of the method. Needless to say, these vectors are rarely known.
- If we follow (36), add "smart zero" to (30), and apply SPPM to the resulting formulation, we automatically get a "star" variant of SPPM-AS, and Theorem 5 captures the complexity of this method.

*Proof of Theorem 5.* From Lemma 5 we know that Assumption 7 holds for the iterates of SPPM* (Algorithm 5) with

$$A_1 = 0, B_1 = 0, C_1 = 0, \quad \text{and} \quad A_2 = 0, B_2 = 0, C_2 = 0.$$

Therefore, from Theorem 1, by choosing any $\alpha > 0$, $\theta = \left(\frac{1}{1+\gamma\mu}\right)^2$, $\zeta = 0$ (see (15) and (16)), we have

$$\mathrm{E}\left[\|x_k - x_\star\|^2\right] \leq \left(\frac{1}{1 + \gamma\mu}\right)^{2k} \|x_0 - x_\star\|^2.$$

$\square$

### B.6 STOCHASTIC PROXIMAL POINT METHOD WITH GRADIENT CORRECTION (SPPM-GC)

We consider the stochastic optimization problem (1), i.e.,

$$\min_{x \in \mathbb{R}^d} \{f(x) := \mathrm{E}_{\xi \sim \mathcal{D}}[f_\xi(x)]\},$$

and rely on Assumption 1 (differentiability of $f_\xi$) and Assumption 2 ($\mu$-strong convexity of $f_\xi$). Recall that this implies strong convexity of $f$. Hence $f$ has a unique minimizer, which we denote $x_\star$.

We have already described the SPPM* method – this a variant of SPPM without the neighborhood term in the convergence bound. In order to run it, we need to know $\nabla f_\xi(x_\star)$ for all $\xi$, which is of course something we almost never know; one exception to this is the *interpolation regime*, defined by the assumption that $\nabla f_\xi(x_\star) = 0$ for all $\xi$.

In this section we describe a practical method inspired by SPPM*. The method is based on the following ideas:

- While we do not know $\nabla f_\xi(x_\star)$, what if this quantity could be in some appropriate/useful sense approximated by some vector $g_\xi$ we *do* know?
- One option is to require a quantity such as

$$\|g_\xi - \nabla f_\xi(x_\star)\|^2 \quad \text{or} \quad \mathrm{E}_{\xi \sim \mathcal{D}}\left[\|g_\xi - \nabla f_\xi(x_\star)\|^2\right]$$

    to be "small" in some sense. However, where can such vectors come from? And what should "small" mean?

- One idea is to make an extra assumption on the functions $f_\xi$ that would somehow automatically guarantee the existence and availability of such vectors. So, we trade off easy availability of vectors $g_\xi$ for a limitation on the class of problems we solve this way.

Recall the Similarity Assumption 5. There exists $\delta \geq 0$ such that

$$\mathrm{E}_{\xi \sim \mathcal{D}} \left[ \|\nabla f_\xi(x) - \nabla f(x) - \nabla f_\xi(x_\star)\|^2 \right] \leq \delta^2 \|x - x_\star\|^2, \qquad \forall x \in \mathbb{R}^d.$$

The above assumption says that we can consider the vectors $g_\xi = \nabla f_\xi(x) - \nabla f(x)$ for any $x$, and that by "small enough" we require a bound by $\delta^2 \|x - x_\star\|^2$. Why these particular choices make sense will become clear from the convergence proof. Let us now make some observations about the class of functions satisfying Assumption 5:

1. The approximation of $\nabla f_\xi(x_\star)$ by $\nabla f_\xi(x) - \nabla f(x)$ gets better as $x$ gets closer to $x_\star$.

2. The smaller $\delta$ is, the better the approximation.

3. If $\delta = 0$, we get perfect approximation.

4. Since
$$\mathrm{E}_{\xi \sim \mathcal{D}} \left[ \nabla f_\xi(x) - \nabla f_\xi(x_\star) \right] = \nabla f(x),$$
the left-hand side of (5) is the variance of the random vector $\nabla f_\xi(x) - \nabla f_\xi(x_\star)$ as an estimator of $\nabla f(x) = \nabla f(x) - \nabla f(x_\star)$.

5. It follows that (5) can be equivalently written in the form
$$\mathrm{E}_{\xi \sim \mathcal{D}} \left[ \|\nabla f_\xi(x) - \nabla f_\xi(x_\star)\|^2 \right] - \|\nabla f(x) - \nabla f(x_\star)\|^2 \leq \delta^2 \|x - x_\star\|^2, \qquad \forall x \in \mathbb{R}^d. \tag{39}$$

6. Note that (39) holds if the following stronger condition holds:
$$\mathrm{E}_{\xi \sim \mathcal{D}} \left[ \|\nabla f_\xi(x) - \nabla f_\xi(x_\star)\|^2 \right] \leq \delta^2 \|x - x_\star\|^2, \qquad \forall x \in \mathbb{R}^d. \tag{40}$$

7. Furthermore, (40) holds if the following even stronger condition holds: there exists $\delta \geq 0$ such that
$$\|\nabla f_\xi(x) - \nabla f_\xi(x_\star)\| \leq \delta \|x - x_\star\|, \qquad \forall x \in \mathbb{R}^d \tag{41}$$
for all $\xi$.

8. Finally, (41) holds if there exists $\delta \geq 0$ such that $\nabla f_\xi$ is $\delta$-Lipschitz for all $\xi$:
$$\|\nabla f_\xi(x) - \nabla f_\xi(y)\| \leq \delta \|x - y\|, \qquad \forall x, y \in \mathbb{R}^d. \tag{42}$$

For each $\xi \in [n]$ and any "parameter" $v \in \mathbb{R}^d$, define

$$\tilde{f}_\xi^v(x) := f_\xi(x) - \langle \nabla f_\xi(v) - \nabla f(v), x \rangle, \tag{43}$$

and instead of solving (1), consider solving the problem

$$\min_{x \in \mathbb{R}^d} \left\{ \tilde{f}^v(x) := \mathrm{E}_{\xi \sim \mathcal{D}} \left[ \tilde{f}_\xi^v(x) \right] \right\}. \tag{44}$$

---

**Algorithm 6** Stochastic Proximal Point Method with Gradient Correction (SPPM-GC)

1: **Parameters:** learning rate $\gamma > 0$, starting point $x_0 \in \mathbb{R}^d$
2: **for** $k = 0, 1, 2, \ldots$ **do**
3:     Sample $\xi_k \sim \mathcal{D}$
4:     $h_k = \nabla f_{\xi_k}(x_k) - \nabla f(x_k)$
5:     $x_{k+1} = \mathrm{prox}_{\gamma f_{\xi_k}} (x_k + \gamma h_k)$                 $= \mathrm{prox}_{\gamma \tilde{f}_{\xi_k}^{x_k}} (x_k)$
6: **end for**

---

The algorithm can be interpreted as:

- a practical variant of SPPM* in which we use the computable correction $h_k = \nabla f_{\xi_k}(x_k) - \nabla f(x_k)$ instead of incomputable correction $\nabla f_{\xi_k}(x_\star)$,

- SPPM applied to the reformulated problem (44), with the "control" vector $v = x_k$ at iteration $k$:

$$v = x_k, \qquad x_{k+1} = \text{prox}_{\gamma \tilde{f}_{\xi_k}^v}(x_k)$$

**Hiding the prox.** Further, recall that if for some $x \in \mathbb{R}^d$ and differentiable and convex $\phi$ we let $x_+ := \text{prox}_\phi(x)$, then $x_+ = x - \nabla\phi(x_+)$. Therefore, steps 4 and 5 of the method can be written in the equivalent form

$$x_{k+1} = x_k + \gamma h_k - \gamma \nabla f_{\xi_k}(x_{k+1}) = x_k - \gamma \left( \nabla f_{\xi_k}(x_{k+1}) - \nabla f_{\xi_k}(x_k) + \nabla f(x_k) \right).$$

**Lemma 6** (SPPM-GC). *Suppose Assumption 5 holds with $\delta > 0$. Assumption 7 holds for the iterates of* SPPM-GC *(Algorithm 6) with*

$$A_1 = \delta^2, B_1 = 0, C_1 = 0, \quad and \quad A_2 = 0, B_2 = 0, C_2 = 0.$$

*Proof.* Recall that the iterates of SPPM-GC have the form

$$x_{k+1} = \text{prox}_{\gamma f_{\xi_k}}(x_k + \gamma h_k),$$

where $h_k = \nabla f_{\xi_k}(x_k) - \nabla f(x_k)$. Let $\sigma^2 \equiv 0$. Since Assumption 5 holds, we get

$$\mathrm{E}\left[ \left. \|h_k - \nabla f_{\xi_k}(x_\star)\|^2 \right| x_k, \sigma_k^2 \right] = \mathrm{E}\left[ \left. \|\nabla f_{\xi_k}(x_k) - \nabla f(x_k) - \nabla f_{\xi_k}(x_\star)\|^2 \right| x_k \right] \leq \delta^2 \|x_k - x_\star\|^2.$$

Hence, Assumption 7 holds with

$$A_1 = \delta^2, B_1 = 0, C_1 = 0, \quad and \quad A_2 = 0, B_2 = 0, C_2 = 0.$$

$\square$

**Theorem 6.** *Let Assumption 1 (differentiability), Assumption 2 ($\mu$-strong convexity), and Assumption 5 ($\delta$-similarity) hold. Choose any $x_0 \in \mathbb{R}^d$. Then for any $\gamma > 0$, and all $k \geq 0$, we have*

$$\mathrm{E}\left[ \|x_k - x_\star\|^2 \right] \leq \left( \frac{1 + \gamma^2\delta^2}{(1 + \gamma\mu)^2} \right)^k \|x_0 - x_\star\|^2. \tag{45}$$

Commentary:

1. **Perfect similarity.** If $\delta = 0$, for all $\gamma > 0$ we get

$$\mathrm{E}\left[ \|x_k - x_\star\|^2 \right] \leq \left( \frac{1}{1 + \gamma\mu} \right)^{2k} \|x_0 - x_\star\|^2. \tag{46}$$

   We get convergence even with $k = 1$ provided that $\gamma$ is chosen large enough. This rate is identical to what Theorem 2 predicts in the interpolation regime. However, the methods are different, and we do not need to assume interpolation regime here. Instead, we assume perfect similarity ($\delta = 0$), and the availability of the gradient of $f$. So, while the rates are exactly the same, both the methods and the assumptions are different.

2. **General case.** It can be shown that the expression $\frac{1 + \gamma^2\delta^2}{(1 + \gamma\mu)^2}$ is minimized for $\gamma = \frac{\mu}{\delta^2}$. With this choice of the stepsize we get

$$\mathrm{E}\left[ \|x_k - x_\star\|^2 \right] \leq \left( \frac{\delta^2}{\delta^2 + \mu^2} \right)^k \|x_0 - x_\star\|^2 = \left( 1 - \frac{\mu^2}{\delta^2 + \mu^2} \right)^k \|x_0 - x_\star\|^2. \tag{47}$$

   This means that

$$k \geq \left( 1 + \frac{\delta^2}{\mu^2} \right) \log\left( \frac{\|x_0 - x_\star\|^2}{\varepsilon} \right) \tag{48}$$

   iterations suffice to guarantee $\mathrm{E}\left[ \|x_k - x_\star\|^2 \right] \leq \varepsilon$.

*Proof of Theorem 6.* From Lemma 6 we have that Assumption 7 holds for the iterates of SPPM-GC (Algorithm 6) with

$$A_1 = \delta^2, B_1 = 0, C_1 = 0, \quad \text{and} \quad A_2 = 0, B_2 = 0, C_2 = 0.$$

From Theorem 1, choosing any $\alpha > 0$, $\theta = \frac{1+\gamma^2\delta^2}{(1+\gamma\mu)^2}$, $\zeta = 0$ (see (15) and (16)), we have

$$\mathrm{E}\left[\|x_k - x_\star\|^2\right] \leq \left(\frac{1+\gamma^2\delta^2}{(1+\gamma\mu)^2}\right)^k \|x_0 - x_\star\|^2.$$

$\square$

## B.7 LOOPLESS STOCHASTIC VARIANCE REDUCED PROXIMAL POINT METHOD (L-SVRP / SPPM-LGC)

We consider the stochastic optimization problem (1), i.e.,

$$\min_{x\in\mathbb{R}^d} \{f(x) := \mathrm{E}_{\xi\sim\mathcal{D}}[f_\xi(x)]\},$$

and rely on Assumption 1 (differentiability of $f_\xi$) and Assumption 2 ($\mu$-strong convexity of $f_\xi$). Recall that this implies strong convexity of $f$. Hence $f$ has a unique minimizer, which we denote $x_\star$.

Note that SPPM-GC needs to compute $\nabla f(x_k)$ in iteration $k$. This can be very costly or even impossible to do in practice. To make this more clear, consider the problem

$$\min_{x\in\mathbb{R}^d} \left\{f(x) = \frac{1}{n}\sum_{i=1}^n f_i(x)\right\}$$

as a special case of (1).

- **One worker.** Assume we have a single machine solving this problem. Moreover, assume it takes one unit of time to this machine to compute $\nabla f_i$ for any $i$, and $n$ units of time to compute $\nabla f$. If the computation of $\nabla f$ is the bottleneck (i.e., if it is more expensive than the evaluation of the proximity operator of $f_i$), then an attempt to design a method addressing this bottleneck would be justified.

- **Parallel workers.** Assume we have $n$ workers able to work in parallel. Then $\nabla f$ can be computed in 1 unit of time if communication among the workers is instantaneous. However, it may still be desirable to avoid having to compute the gradient:

  - The server aggregating the $n$ gradients computed by the workers may have limited capacity, and it make take more time for it to be able to compute the average of a very large number of vectors.
  - Some workers may be not available at all times.

These considerations justify the desire to reduce the reliance of SPPM-GC on the computation of $\nabla f$. The key idea is to compute the gradient only periodically, i.e., to be "lazy" about computing the gradient. In particular, we flip a biased coin, and compute a new gradient if the coin lands the right way. Otherwise, we use the previously computed gradient instead. We shall formalize this in the next section.

We are now ready to present the stochastic proximal point method with lazy gradient correction (SPPM-LGC). In the literature, the method is known under the name loopless stochastic variance reduced proximal (L-SVRP) point method.

---

**Algorithm 7** Loopless Stochastic Variance Reduced Proximal Point Method (L-SVRP / SPPM-LGC)

---

1: **Parameters:** learning rate $\gamma > 0$, starting point $x_0 \in \mathbb{R}^d$, starting control vector $w_0 \in \mathbb{R}^d$, probability $p \in (0, 1]$
2: **for** $k = 0, 1, 2, \ldots$ **do**
3:      Sample $\xi_k \sim \mathcal{D}$
4:      Set $h_k = \nabla f_{\xi_k}(w_k) - \nabla f(w_k)$
5:      $x_{k+1} = \mathrm{prox}_{\gamma f_{\xi_k}}(x_k + \gamma h_k)$
6:      Set $w_{k+1} = \begin{cases} x_{k+1} & \text{with probability} \quad p \\ w_k & \text{with probability} \quad 1 - p \end{cases}$
7: **end for**

---

Note that as intended, L-SVRP indeed reduces to SPPM-GC when $w_0 = x_0$ and $p = 1$.

**Hiding the prox.** Further, recall that if for some $x \in \mathbb{R}^d$ and differentiable and convex $\phi$ we let $x_+ := \mathrm{prox}_\phi(x)$, then $x_+ = x - \nabla\phi(x_+)$. Therefore, steps 4 and 5 of the method can be written in the equivalent form

$$x_{k+1} = x_k + \gamma h_k - \gamma \nabla f_{\xi_k}(x_{k+1}) = x_k - \gamma \left( \nabla f_{\xi_k}(x_{k+1}) - \nabla f_{\xi_k}(w_k) + \nabla f(w_k) \right).$$

**L-SVRP vs L-SVRG.** The name L-SVRP was intentionally coined to resemble the name L-SVRG, which is a method proposed in and studied by Kovalev et al. (2020). This method has the form

$$x_{k+1} = x_k - \gamma \left( \nabla f_{\xi_k}(x_k) - \nabla f_{\xi_k}(w_k) + \nabla f(w_k) \right),$$

with Step 6 being identical. That is, the only difference here is that L-SVRP involves $\nabla f_{\xi_k}(x_{k+1})$ while L-SVRG uses $\nabla f_{\xi_k}(x_k)$ in the same place. So, "P" in L-SVRP refers to the proximal nature of the term $\nabla f_{\xi_k}(x_{k+1})$, while "G" in L-SVRG refers to the gradient nature of the corresponding term $\nabla f_{\xi_k}(x_k)$.

**Loopless vs loopy structure.** The word "loopless" refers to the way the control vector $w_{k+1}$ is updated in Step 6. The alternative to this, used in the famous SVRG method of Johnson and Zhang (2013), is to update $w_{k+1}$ once every $m$ iterations, where $m$ is an appropriately chosen parameter. This change introduces an outer loop into the method, and makes it look a bit more cumbersome. More importantly, the loopless nature of L-SVRG is useful in three ways:

(i) leads to a somewhat sharper analysis,

(ii) makes the method easier to analyze, and

(iii) allows for easier to extensions / modifications.

The last two points are more important than the first one.

**Lemma 7** (L-SVRP). *Suppose Assumption 5 holds with $\delta > 0$. Assumption 7 holds for the iterates of L-SVRP (Algorithm 7) with*

$$A_1 = 0, B_1 = \delta^2, C_1 = 0, \quad and \quad A_2 = p, B_2 = 1 - p, C_2 = 0.$$

*Proof of Lemma 7.* Recall that the iterates of L-SVRP have the form

$$x_{k+1} = \mathrm{prox}_{\gamma f_{\xi_k}}(x_k + \gamma h_k),$$

where $h_k$ is defined as $h_k = \nabla f_{\xi_k}(w_k) - \nabla f(w_k)$, and $w_k$ is updated in a loopless fashion. Let $\phi_k = w_k$. Then

$$\mathrm{E}\left[h_k | x_k, \phi_k\right] = \mathrm{E}\left[\nabla f_{\xi_k}(w_k) - \nabla f(w_k) | x_k, w_k\right] = 0,$$

and hence (7) holds. If, moreover, Assumption 5 holds, then

$$\mathrm{E}\left[\left.\|h_k - \nabla f_{\xi_k}(x_\star)\|^2\right| x_k, w_k\right] \overset{(80)}{\leq} \delta^2 \|w_k - x_\star\|^2,$$

which means that (8) holds with $A_1 = 0$, $B_1 = \delta^2$ and $C_1 = 0$ if we let $\sigma^2(z) := \|z - x_\star\|^2$ (since then $\sigma_k^2 = \sigma^2(w_k) = \|w_k - x_\star\|^2$). On the other hand, from the proof of auxiliary Lemma 12 we know that

$$\mathrm{E}\left[\|w_{k+1} - x_\star\|^2 \,\Big|\, x_{k+1}, w_k\right] \overset{(81)}{=} p\|x_{k+1} - x_\star\|^2 + (1 - p)\|w_k - x_\star\|^2,$$

which means that (9) holds with $A_2 = p$, $B_2 = 1 - p$, and $C_2 = 0$. In summary, Assumption 7 holds with

$$A_1 = 0, B_1 = \delta^2, C_1 = 0, \quad \text{and} \quad A_2 = p, B_2 = 1 - p, C_2 = 0.$$

$\square$

**Theorem 7.** *Let Assumption 1 (differentiability), Assumption 2 ($\mu$-strong convexity), and Assumption 5 ($\delta$-similarity) hold. Choose any $x_0, w_0 \in \mathbb{R}^d$. Then for any $p \in (0, 1]$, $\gamma > 0$, $\alpha > 0$ and all $k \geq 0$, we have*

$$\mathrm{E}\left[\Psi_k\right] \leq \max\left\{\frac{1 + \alpha p}{(1 + \gamma\mu)^2}, \frac{1 + \alpha p}{(1 + \gamma\mu)^2}\frac{\gamma^2\delta^2}{\alpha} + 1 - p\right\}^k \Psi_0, \tag{49}$$

*where*

$$\Psi_k := \|x_k - x_\star\|^2 + \alpha\|w_k - x_\star\|^2. \tag{50}$$

*If $\delta = 0$, we have the more precise result*

$$\mathrm{E}\left[\|x_{k+1} - x_\star\|^2\right] \leq \frac{1}{(1 + \gamma\mu)^2}\mathrm{E}\left[\|x_k - x_\star\|^2\right]. \tag{51}$$

Commentary:

1. **Convergence vs divergence.** Clearly, it is possible for the maximum in (49) to *not* be smaller than 1. In this case, the theorem gives a meaningless result. Whether or not the value is smaller than 1 depends on the choice of the parameters $\alpha$, $p$ and $\gamma$ in relation to the strong convexity constant $\mu$. For example, it's clear that if $\gamma$ and $p$ are fixed and $\alpha$ is too large, the expression $\frac{1+\alpha p}{(1+\gamma\mu)^2}$ might exceed 1, rendering the rate vacuous.

2. **Optimal choice of $\alpha$.** Note that $\alpha \mapsto \frac{1+\alpha p}{(1+\gamma\mu)^2}$ is linear and increasing, and $\alpha \mapsto \frac{1+\alpha p}{(1+\gamma\mu)^2}\frac{\gamma^2\delta^2}{\alpha} + 1 - p$ is convex and decreasing (make sure you understand why!). Moreover, while the first function has a finite value at $\alpha = 0$, the second function blows up as $\alpha$ approaches 0 from the right. This means that the maximum of these two functions will be minimized at the point where the graphs of the two functions intersect, i.e., at $\alpha$ satisfying

$$\frac{1 + \alpha p}{(1 + \gamma\mu)^2} = \frac{1 + \alpha p}{(1 + \gamma\mu)^2}\frac{\gamma^2\delta^2}{\alpha} + 1 - p. \tag{52}$$

In the $p = 1$ case (L-SVRP reduces to SPPM-GC in this regime), the equation simplifies to $1 = \frac{\gamma^2\delta^2}{\alpha}$, i.e., the optimal solution is $\alpha = \gamma^2\delta^2$, and (49) reduces to

$$\mathrm{E}\left[\Psi_k\right] \leq \left(\frac{1 + \gamma^2\delta^2}{(1 + \gamma\mu)^2}\right)^k \Psi_0. \tag{53}$$

This is the same result we obtained in (47) for the SPPM-GC method, up to the choice of the Lyapunov function. However, if we initialize with $w_0 = x_0$, then

$$\Psi_k = (1 + \gamma\mu)\|x_k - x_\star\|^2, \tag{54}$$

and plugging this into (53) gives

$$\mathrm{E}\left[\|x_k - x_\star\|^2\right] \leq \left(\frac{1 + \gamma^2\delta^2}{(1 + \gamma\mu)^2}\right)^k \|x_0 - x_\star\|^2, \tag{55}$$

which is exactly (47).

The $0 < p < 1$ case turns out to be a more cumbersome. After a bit of algebra, one obtains that equation (52) is equivalent to the quadratic equation

$$p\alpha^2 + b\alpha + c = 0,$$

where $b = 1 - (1 + \gamma\mu)^2(1 - p) - p\gamma^2\delta^2$ and $c = \gamma^2\delta^2$. The roots of the quadratic are

$$\alpha = \frac{-b \pm \sqrt{b^2 - 4pc}}{2p}.$$

It seems it is awkward to work with this expression since the resulting rate will become hard to parse and interpret. So, we'll give up on working with perfectly optimal $\alpha$. Nevertheless, we will show how to choose some (slightly suboptimal) $\alpha$ in Corollary 8 which also gives the right complexity result.

Admittedly, it's not easy to understand how good is the rate provided by (49). The following corollary sheds light on what is achievable.

**Corollary 8.** *If we choose $\alpha = \frac{\gamma\mu}{p}$ and $\gamma = \frac{p}{p\frac{\delta^2}{\mu} + (1-p)\mu}$, then for any $\varepsilon > 0$ we have*

$$k \geq \left(\frac{1}{p} + \frac{\delta^2}{\mu^2}\right)\log\left(\frac{\Psi_0}{\varepsilon}\right) \qquad \Rightarrow \qquad \mathrm{E}\left[\Psi_k\right] \leq \varepsilon. \tag{56}$$

*Proof.* Since $\alpha = \frac{\gamma\mu}{p}$, we have

$$A(\gamma) := \frac{1 + \alpha p}{(1 + \gamma\mu)^2} = \frac{1}{1 + \gamma\mu}$$

and

$$B(\gamma) := \frac{1 + \alpha p}{(1 + \gamma\mu)^2}\frac{\gamma^2\delta^2}{\alpha} + 1 - p = \frac{1}{1 + \gamma\mu}\frac{p\gamma\delta^2}{\mu} + 1 - p.$$

Plugging this into (50) leads to

$$\mathrm{E}\left[\Psi_k\right] \leq \max\left\{A(\gamma), B(\gamma)\right\}^k \Psi_0, \tag{57}$$

We will now select stepsize $\gamma$ which minimizes

$$\gamma \mapsto \max\{A(\gamma), B(\gamma)\}.$$

Notice that $\gamma \mapsto A(\gamma)$ is decreasing to zero on $(0, \infty)$, with $A(\gamma)$ blowing up to $\infty$ as $\gamma$ approaches zero from the right. Further, $\gamma \mapsto B(\gamma)$ is increasing in $(0, \infty)$. This means that $\max\{A(\gamma), B(\gamma)\}$ is minimized at the point where the graphs of the two functions intersect, i.e., at $\gamma$ satisfying $A(\gamma) = B(\gamma)$. Direct calculation shows that the solution of this is

$$\gamma = \gamma_\star := \frac{p}{p\frac{\delta^2}{\mu} + (1-p)\mu}, \tag{58}$$

and hence

$$\mathrm{E}\left[\Psi_k\right] \leq \max\left\{A(\gamma_\star), B(\gamma_\star)\right\}^k \Psi_0 = A(\gamma_\star)^k \Psi_0 = \left(\frac{1}{1 + \gamma_\star\mu}\right)^k \Psi_0 = \left(1 - \frac{\gamma_\star\mu}{1 + \gamma_\star\mu}\right)^k \Psi_0.$$

This implies that

$$k \geq \left(1 + \frac{1}{\gamma_\star\mu}\right)\log\left(\frac{\Psi_0}{\varepsilon}\right) \qquad \Rightarrow \qquad \mathrm{E}\left[\Psi_k\right] \leq \varepsilon. \tag{59}$$

Plugging $\gamma_\star$ into this iteration complexity result gives

$$1 + \frac{1}{\gamma_\star\mu} \overset{(58)}{=} 1 + \frac{\frac{p\delta^2}{\mu} + (1-p)\mu}{p\mu} = \frac{1}{p} + \frac{\delta^2}{\mu^2}.$$

Plugging this back into (59) gives the final result

$$k \geq \left(\frac{1}{p} + \frac{\delta^2}{\mu^2}\right)\log\left(\frac{\Psi_0}{\varepsilon}\right) \qquad \Rightarrow \qquad \mathrm{E}\left[\Psi_k\right] \leq \varepsilon. \tag{60}$$

$\square$

Commentary:

1. **Comparing to SPPM with Gradient Correction (i.e., $p = 1$).** Recall that L-SVRP reduces to SPPM-GC when $w_0 = x_0$ and $p = 1$. Therefore, one would expect the rates to be the same. First, notice that $w_k = x_k$ for all $k$, and as a result, the Lyapunov function (50) reduces to

$$\Psi_k = (1 + \gamma\mu)\|x_k - x_\star\|^2, \tag{61}$$

and hence (57) reduces to

$$(1 + \gamma\mu)\mathrm{E}\left[\|x_k - x_\star\|^2\right] \overset{(61)}{=} \mathrm{E}\left[\Psi_k\right] \leq \max\left\{\left(\frac{1}{1+\gamma\mu}\right)^k, \left(\frac{1}{1+\gamma\mu}\frac{\gamma\delta^2}{\mu}\right)^k\right\}\Psi_0.$$

If we choose $\gamma \leq \frac{\mu}{\delta^2}$, then the first term in the max dominates, and we get

$$\max\left\{\left(\frac{1}{1+\gamma\mu}\right)^k, \left(\frac{1}{1+\gamma\mu}\frac{\gamma\delta^2}{\mu}\right)^k\right\}\Psi_0 \quad = \quad \left(\frac{1}{1+\gamma\mu}\right)^k\Psi_0$$

$$\overset{(61)}{=} \quad \left(\frac{1}{1+\gamma\mu}\right)^k (1+\gamma\mu)\|x_0 - x_\star\|^2.$$

Combining the above observations, we get

$$\mathrm{E}\left[\|x_k - x_\star\|^2\right] \leq \left(\frac{1}{1+\gamma\mu}\right)^k \|x_0 - x_\star\|^2 = \left(1 - \frac{1}{1+\frac{1}{\gamma\mu}}\right)^k \|x_0 - x_\star\|^2.$$

It is easy to see that this means that if we choose $k \geq \left(1 + \frac{1}{\gamma\mu}\right)\log\left(\frac{\|x_0 - x_\star\|^2}{\varepsilon}\right)$, then $\mathrm{E}\left[\|x_k - x_\star\|^2\right] \leq \varepsilon$. The best rate is obtained for the largest allowed stepsize, i.e., for $\gamma = \frac{\mu}{\delta^2}$ (recall that this was the optimal stepsize choice for SPPM-GC established in Appendix B.6), in which case we conclude that

$$k \geq \left(1 + \frac{\delta^2}{\mu^2}\right)\log\left(\frac{\|x_0 - x_\star\|^2}{\varepsilon}\right) \quad \Rightarrow \quad \mathrm{E}\left[\|x_k - x_\star\|^2\right] \leq \varepsilon.$$

If more similarity (i.e., smaller $\delta$) or more strong convexity (i.e., larger $\mu$) is present, fewer iterations are needed to solve the problem. Note that we get the same result as in (48); so, we do not lose anything by doing the analysis in the $\delta > 0$ case using the Lyapunov approach.

2. **Comparison to the result of Khaled and Jin (2023).** Choosing $\alpha = \frac{\gamma\mu}{p}, \gamma = \frac{\mu}{2\delta^2}, p = \frac{1}{n}$, we retrieve the convergence guarantees of Khaled and Jin (2023). Indeed, from Corollary 8 we have that

$$A(\gamma) = \frac{1}{1+\gamma\mu}, \quad B(\gamma) = \frac{1}{1+\gamma\mu}\frac{\gamma\delta^2 p}{\mu} + 1 - p.$$

The condition on the stepsize states that $\frac{\gamma\delta^2}{\mu} \leq \frac{1}{2}$. It implies that $B(\gamma) \leq 1 - \frac{p}{2}$. Let $\rho = \min\left\{\frac{\gamma\mu}{1+\gamma\mu}, \frac{p}{2}\right\}$. Clearly, $\mathrm{E}\left[\|x_k - x_\star\|^2\right] \leq \mathrm{E}\left[\Psi_k\right]$. Further, as $w_0 = x_0$,

$$\mathrm{E}\left[\Psi_0\right] = \|x_0 - x_\star\|^2 + \frac{\gamma\mu}{p}\|w_0 - x_\star\|^2 = \left(1 + \frac{\gamma\mu}{p}\right)\|x_0 - x_\star\|^2.$$

For any $k \geq 0$, we obtain

$$\mathrm{E}\left[\|x_k - x_\star\|^2\right] \leq \left(1 + \frac{\gamma\mu}{p}\right)(1 - \rho)^k \|x_0 - x_\star\|^2,$$

therefore,

$$\mathrm{E}\left[\|x_K - x_\star\|^2\right] \leq \left(1 + \frac{\gamma\mu}{p}\right)\exp\left\{-\rho K\right\}\|x_0 - x_\star\|^2.$$

If we run L-SVRP for

$$K \geq \frac{1}{\rho} \log \left( \frac{\|x_0 - x_\star\|^2 \left(1 + \frac{\gamma\mu}{p}\right)}{\varepsilon} \right).$$

Making the substitutions $\rho = \min\left\{\frac{\gamma\mu}{1+\gamma\mu}, \frac{p}{2}\right\}, \gamma = \frac{\mu}{2\delta^2}, p = \frac{1}{n}$, we arrive at

$$K \geq 2 \max\left\{\frac{1}{2} + \frac{\delta^2}{\mu^2}, n\right\} \log \left( \frac{\|x_0 - x_\star\|^2 \left(1 + \frac{\mu^2 n}{2\delta^2}\right)}{\varepsilon} \right),$$

which is even slightly better than the result by Khaled and Jin (2023).

*Proof of Theorem 7.* From Lemma 7 we know that Assumption 7 holds for the iterates of L-SVRP (Algorithm 7) with

$$A_1 = 0, B_1 = \delta^2, C_1 = 0, \quad \text{and} \quad A_2 = p, B_2 = 1 - p, C_2 = 0.$$

From Theorem 1 choosing any $\alpha > 0, \theta = \max\left\{\frac{1+\alpha p}{(1+\gamma\mu)^2}, \frac{1+\alpha p}{(1+\gamma\mu)^2}\frac{\gamma^2\delta^2}{\alpha} + 1 - p\right\}, \zeta = 0$ (see (15) and (16)), we get

$$\mathrm{E}\left[\Psi_k\right] \leq \max\left\{\frac{1+\alpha p}{(1+\gamma\mu)^2}, \frac{1+\alpha p}{(1+\gamma\mu)^2}\frac{\gamma^2\delta^2}{\alpha} + 1 - p\right\}^k \Psi_0,$$

where

$$\Psi_k := \|x_k - x_\star\|^2 + \alpha\|w_k - x_\star\|^2.$$

$\square$

## B.8 POINT SAGA (Point SAGA)

The main motivation is to give one more example of a SPPM method based on the idea of gradient correction which does not need to compute the full/exact gradient of $f$ in each iteration. We consider another well-known variance-reduced SPPM method called Point SAGA. However, we will revert back to the finite-sum optimiziation problem

$$\min_{x \in \mathbb{R}^d} \left\{ f(x) = \frac{1}{n} \sum_{i=1}^{n} f_i(x) \right\}.$$

---

**Algorithm 8** Point SAGA (Point SAGA)

1: **Parameters:** learning rate $\gamma > 0$, starting point $x_0 \in \mathbb{R}^d$, starting control vectors $w_0^i \in \mathbb{R}^d$ for $i \in [n]$
2: **for** $k = 0, 1, 2, \ldots$ **do**
3:     Sample $i_k \in \{1, \ldots, n\}$ uniformly at random
4:     Set $h_k = \nabla f_{i_k}(w_k^{i_k}) - \frac{1}{n}\sum_{j=1}^{n} \nabla f_j(w_k^j)$
5:     $x_{k+1} = \mathrm{prox}_{\gamma f_{i_k}}(x_k + \gamma h_k)$
6:     Set $w_{k+1}^j = \begin{cases} x_{k+1} & \text{for} \quad j = i_k \\ w_k^j & \text{for} \quad j \neq i_k \end{cases}$
7: **end for**

---

Commentary:

1. Compared to Algorithm 7, Algorithm 8 uses additional memory to store the table of control vectors $w_k^i$ or computed gradients $\nabla f_i(w_k^i)$.

2. In each iteration, the method update only one "column" in a memory table replacing the old control vector/gradient with the corresponding new one.

The structure of Point SAGA will not allow us perform the analysis under the similarity assumption (Assumption 5). Instead, we will rely on the stronger Assumption 6.

We assume that there exists $\nu > 0$ such that the inequality

$$\frac{1}{n} \sum_{j=1}^{n} \left\| \nabla f_j(x^j) - \frac{1}{n} \sum_{i=1}^{n} \nabla f_i(x^i) - \nabla f_j(x_\star) \right\|^2 \leq \nu^2 \frac{1}{n} \sum_{j=1}^{n} \left\| x^j - x_\star \right\|^2$$

holds for all $x^1, \ldots, x^n \in \mathbb{R}^d$.

This inequality can be written in the form

$$\frac{1}{n} \sum_{j=1}^{n} \left\| \nabla f_j(x^j) - \nabla f_j(x_\star) \right\|^2 - \left\| \frac{1}{n} \sum_{i=1}^{n} \nabla f_i(x^i) \right\|^2 \leq \nu^2 \frac{1}{n} \sum_{j=1}^{n} \left\| x^j - x_\star \right\|^2, \quad \forall x^j \in \mathbb{R}^d, \ j \in [n]. \tag{62}$$

Thus, (62) holds, if the following condition is assumed

$$\frac{1}{n} \sum_{j=1}^{n} \left\| \nabla f_j(x^j) - \nabla f_j(x_\star) \right\|^2 \leq \nu^2 \frac{1}{n} \sum_{j=1}^{n} \left\| x^j - x_\star \right\|^2, \quad \forall x^j \in \mathbb{R}^d, \ j \in [n]. \tag{63}$$

Moreover, (63) is equivalent to the following condition: for all $j \in [n]$, we have

$$\left\| \nabla f_j(x) - \nabla f_j(x_\star) \right\|^2 \leq \nu^2 \|x - x_\star\|^2, \quad \forall x \in \mathbb{R}^d. \tag{64}$$

Finally, (64) holds, if each $f_j$ is $\nu$-smooth, i.e.

$$\left\| \nabla f_j(x) - \nabla f_j(y) \right\|^2 \leq \nu^2 \|x - y\|^2, \quad \forall x, y \in \mathbb{R}^d. \tag{65}$$

In summary, we have the following relations between the above conditions:

$$(65) \quad \Rightarrow \quad (64) \quad \equiv \quad (63) \quad \Rightarrow \quad (62) \quad \equiv \quad (6).$$

**Lemma 8.** *Suppose Assumption 6 holds with $\nu > 0$. Then Assumption 7 holds for the iterates of* Point SAGA *(Algorithm 8) with*

$$A_1 = 0, B_1 = \nu^2, C_1 = 0, \quad and \quad A_2 = \frac{1}{n}, B_2 = \frac{n-1}{n}, C_2 = 0.$$

*Proof of Lemma 8.* Let $\sigma_k = \frac{1}{n} \sum_{i=1}^{n} \left\| w_k^i - x_\star \right\|^2$, $\phi_k = \left( w_k^1, \ldots, w_k^n \right)$. Recalling that

$$h_k := \nabla f_{i_k}(w_k^{i_k}) - \frac{1}{n} \sum_{j=1}^{n} \nabla f_j(w_k^j), \tag{66}$$

we have

$$\mathrm{E}\left[ h_k \, \middle| \, x_k, \phi_k \right] \quad = \quad \mathrm{E}\left[ \frac{1}{n} \sum_{j=1}^{n} \nabla f_j(w_k^j) - \frac{1}{n} \sum_{j=1}^{n} \nabla f_j(w_k^j) \, \middle| \, x_k, \phi_k \right] = 0. \tag{67}$$

Further,

$$\mathrm{E}\left[ \left\| h_k - \nabla f_{i_k}(x_\star) \right\|^2 \, \middle| \, x_k, \phi_k \right] \quad \overset{(66)}{=} \quad \mathrm{E}\left[ \left\| \nabla f_{i_k}(w_k^{i_k}) - \frac{1}{n} \sum_{j=1}^{n} \nabla f_j(w_k^j) - \nabla f_{i_k}(x_\star) \right\|^2 \, \middle| \, x_k, \phi_k \right]$$

$$= \quad \frac{1}{n} \sum_{i=1}^{n} \left\| \nabla f_i(w_k^i) - \frac{1}{n} \sum_{j=1}^{n} \nabla f_j(w_k^j) - \nabla f_i(x_\star) \right\|^2$$

$$\overset{(6)}{\leq} \quad \frac{\nu^2}{n} \sum_{i=1}^{n} \left\| w_k^i - x_\star \right\|^2.$$

Therefore, we have that $A_1 = 0$, $B_1 = \nu^2$, $C_1 = 0$.

$$
\begin{aligned}
\mathrm{E}\left[\sigma_{k+1}^2 \,\middle|\, x_{k+1}, \phi_k\right] &= \mathrm{E}\left[\frac{1}{n}\sum_{i=1}^{n}\left\|w_{k+1}^i - x_\star\right\|^2 \,\middle|\, x_{k+1}, \phi_k\right] \\
&= \frac{1}{n}\sum_{i_k=1}^{n}\left[\frac{1}{n}\|x_{k+1} - x_\star\|^2 + \frac{1}{n}\sum_{j\neq i_k}\left\|w_k^j - x_\star\right\|^2\right] \\
&= \frac{1}{n}\|x_{k+1} - x_\star\|^2 + \frac{n-1}{n}\sigma_k^2.
\end{aligned}
$$

Therefore, we have that $A_2 = \frac{1}{n}$, $B_2 = \frac{n-1}{n}$, $C_2 = 0$. $\qquad\square$

The convergence of Point SAGA is captured by the following theorem.

**Theorem 9.** *Let Assumption 1, Assumption 2 and Assumption 6 hold. Chose any $x_0, w_0^1, \ldots, w_0^n \in \mathbb{R}^d$. Then for any $\gamma > 0$, and all $k \geq 0$, we have*

$$
\mathrm{E}\left[\Psi_k\right] \leq \max\left\{\left(\frac{1}{1+\gamma\mu}\right)^k, \left(\frac{1}{1+\gamma\mu}\frac{\gamma\nu^2}{\mu n} + 1 - \frac{1}{n}\right)^k\right\}\Psi_0, \tag{68}
$$

*where*

$$
\Psi_k := \|x_k - x_\star\|^2 + \gamma\mu\sum_{i=1}^{n}\left\|w_k^i - x_\star\right\|^2. \tag{69}
$$

Clearly, it is possible for the maximum in (68) to not be smaller than 1. In this case, the theorem produces the meaningless result. Whether or not the value is smaller than 1 dependes on the choice of $\gamma$ with respect to the strong convexity constant $\mu$, the number of individual functions $n$, the similarity constant $\nu$.

*Proof of Theorem 9.* From Lemma 8 we have that Assumption 7 holds for the iterates of Point SAGA (Algorithm 8) with

$$
A_1 = 0, B_1 = \nu^2, C_1 = 0, \quad \text{and} \quad A_2 = \frac{1}{n}, B_2 = \frac{n-1}{n}, C_2 = 0.
$$

From Theorem 1, choosing $\alpha = \gamma\mu n$, $\theta = \max\left\{\frac{1}{1+\gamma\mu}, \frac{1}{1+\gamma\mu}\frac{\gamma\nu^2}{\mu n} + 1 - \frac{1}{n}\right\}$, $\zeta = 0$ (see (15) and (16)), we obtain

$$
\mathrm{E}\left[\Psi_k\right] \leq \max\left\{\left(\frac{1}{1+\gamma\mu}\right)^k, \left(\frac{1}{1+\gamma\mu}\frac{\gamma\nu^2}{\mu n} + 1 - \frac{1}{n}\right)^k\right\}\Psi_0,
$$

where

$$
\Psi_k := \|x_k - x_\star\|^2 + \gamma\mu\sum_{i=1}^{n}\left\|w_k^i - x_\star\right\|^2.
$$

$\qquad\square$

**Corollary 10.** *If we choose $\gamma = \frac{1}{\frac{\nu^2}{\mu} + (n-1)\mu}$, then, for any $\varepsilon > 0$, we have*

$$
k \geq \left(n + \frac{\nu^2}{\mu^2}\right)\log\left(\frac{\Psi_0}{\varepsilon}\right) \qquad \Rightarrow \qquad \mathrm{E}\left[\Psi_k\right] \leq \varepsilon.
$$

*Proof of Corollary 10.* Notice that $A(\gamma) := \frac{1}{1+\gamma\mu}$ is decreasing for $\gamma > 0$, and $B(\gamma) := \frac{\gamma\nu^2}{1+\gamma\mu} + 1 - \frac{1}{n}$ is increasing for $\gamma > 0$. This means that $\max\{\}$ is minimized at $\gamma := \gamma_\star$ where $A(\gamma) = B(\gamma)$. Direct calculation shows that the solution of this is

$$
\gamma = \gamma_\star := \frac{1}{\frac{\nu^2}{\mu} + (n-1)\mu},
$$

and hence

$$\mathrm{E}\left[\Psi_k\right] \leq \max\left\{A(\gamma_\star), B(\gamma_\star)\right\}^k \Psi_0 = A(\gamma_\star)^k \Psi_0 = \left(\frac{1}{1+\gamma_\star\mu}\right)^k \Psi_0 = \left(1 - \frac{\gamma_\star\mu}{1+\gamma_\star\mu}\right)^k \Psi_0.$$

This implies that

$$k \geq \left(1 + \frac{1}{\gamma_\star\mu}\right)\log\left(\frac{\Psi_0}{\varepsilon}\right) \qquad \Rightarrow \qquad \mathrm{E}\left[\Psi_k\right] \leq \varepsilon.$$

Plugging $\gamma_\star$ into this iteration complexity result gives

$$1 + \frac{1}{\gamma_\star\mu} = 1 + \frac{\frac{\nu^2}{\mu n} + \left(1 - \frac{1}{n}\right)\mu}{\frac{\mu}{n}} = n + \frac{\nu^2}{\mu^2}.$$

We obtain the final result

$$k \geq \left(n + \frac{\delta^2}{\mu^2}\right)\log\left(\frac{\Psi_0}{\varepsilon}\right) \qquad \Rightarrow \qquad \mathrm{E}\left[\Psi_k\right] \leq \varepsilon.$$

$\square$

## C  AUXILIARY LEMMA FOR SPPM-LC

Our main result relies on a single lemma only.

**Lemma 9.** *Let Assumption 1, Assumption 2 and Assumption 7 hold. Then*

$$\mathrm{E}\left[\|x_{k+1} - x_\star\|^2\right] \leq \frac{(1+\gamma^2 A_1)}{(1+\gamma\mu)^2}\mathrm{E}\left[\|x_k - x_\star\|^2\right] + \frac{\gamma^2 B_1}{(1+\gamma\mu)^2}\mathrm{E}\left[\sigma_k^2\right] + \frac{\gamma^2 C_1}{(1+\gamma\mu)^2}. \qquad (70)$$

*Proof.* By combining Fact 1 and Fact 2, we get

$$\|x_{k+1} - x_\star\|^2 \overset{Fact\ 1}{=} \left\|\mathrm{prox}_{\gamma f_{\xi_k}}(x_k + \gamma h_k) - \mathrm{prox}_{\gamma f_{\xi_k}}(x_\star + \gamma\nabla f_{\xi_k}(x_\star))\right\|^2$$

$$\overset{Fact\ 2}{\leq} \frac{1}{(1+\gamma\mu)^2}\|x_k + \gamma h_k - (x_\star + \gamma\nabla f_{\xi_k}(x_\star))\|^2$$

$$= \frac{1}{(1+\gamma\mu)^2}\|x_k - x_\star + \gamma(h_k - \nabla f_{\xi_k}(x_\star))\|^2.$$

Thus, we have that

$$\|x_{k+1} - x_\star\|^2 \leq \frac{1}{(1+\gamma\mu)^2}\left(\|x_k - x_\star\|^2 + 2\gamma\langle h_k - \nabla f_{\xi_k}(x_\star), x_k - x_\star\rangle + \gamma^2\|h_k - \nabla f_{\xi_k}(x_\star)\|^2\right).$$

We can use it since it holds irrespective of the choice of $h_k$. Taking conditional expectation on both sides, we get

$$\mathrm{E}\left[\|x_{k+1} - x_\star\|^2 \,\middle|\, x_k, \phi_k\right] \leq \frac{1}{(1+\gamma\mu)^2}\mathrm{E}\left[\|x_k - x_\star\|^2 \,\middle|\, x_k, \phi_k\right]$$

$$+ \frac{2\gamma}{(1+\gamma\mu)^2}\langle\mathrm{E}\left[h_k - \nabla f_{\xi_k}(x_\star)\,\middle|\, x_k, \phi_k\right], x_k - x_\star\rangle$$

$$+ \frac{\gamma^2}{(1+\gamma\mu)^2}\mathrm{E}\left[\|h_k - \nabla f_{\xi_k}(x_\star)\|^2 \,\middle|\, x_k, \phi_k\right]. \qquad (71)$$

Note that

$$\mathrm{E}\left[\|x_k - x_\star\|^2 \,\middle|\, x_k, \phi_k\right] = \|x_k - x_\star\|^2. \qquad (72)$$

Further,

$$\mathrm{E}\left[h_k - \nabla f_{\xi_k}(x_\star)\,\middle|\, x_k, \phi_k\right] = \mathrm{E}\left[h_k\,\middle|\, x_k, \phi_k\right] - \mathrm{E}\left[\nabla f_{\xi_k}(x_\star)\,\middle|\, x_k, \phi_k\right]$$

$$= \mathrm{E}\left[h_k\,\middle|\, x_k, \phi_k\right] - \nabla f(x_\star)$$

$$= \mathrm{E}\left[h_k\,\middle|\, x_k, \phi_k\right]$$

$$= 0, \qquad (73)$$

where the last equality follows from Assumption 7, relation (7). Relation (8) of Assumption 7 says that

$$\mathrm{E}\left[\left.\|h_k - \nabla f_{\xi_k}(x_\star)\|^2 \right| x_k, \phi_k\right] \leq A_1\|x_k - x_\star\|^2 + B_1\sigma_k^2 + C_1. \tag{74}$$

Plugging (74), (73) and (72) into (71), we obtain

$$\mathrm{E}\left[\left.\|x_{k+1} - x_\star\|^2 \right| x_k, \phi_k\right] \leq \frac{1}{(1+\gamma\mu)^2}\left(\left(1 + \gamma^2 A_1\right)\|x_k - x_\star\|^2 + \gamma^2 B_1\sigma_k^2 + \gamma^2 C_1\right). \tag{75}$$

It only remains to take expectation on both sides and apply the tower property. □

## D  AUXILIARY LEMMAS

### D.1  SPPM-AS

**Lemma 10.** *Let $\phi_1, \ldots, \phi_m : \mathbb{R}^d \to \mathbb{R}$ be differentiable functions, with $\phi_i$ being $\mu_i$-strongly convex for all $i \in [n]$. Further, let $w_1, \ldots, w_m$ be positive scalars. Then the function $\phi := \sum_{i=1}^{m} w_i\phi_i$ is $\mu$-strongly convex with $\mu = \sum_{i=1}^{m} w_i\mu_i$.*

*Proof.* By assumption,

$$\phi_i(y) + \langle \nabla\phi_i(y), x - y \rangle + \frac{\mu_i}{2}\|x - y\|^2 \leq \phi_i(x), \qquad \forall x, y \in \mathbb{R}^d. \tag{76}$$

This means that

$$\sum_{i=1}^{m} w_i\left(\phi_i(y) + \langle \nabla\phi_i(y), x - y \rangle + \frac{\mu_i}{2}\|x - y\|^2\right) \leq \sum_{i=1}^{m} w_i\phi_i(x), \qquad \forall x, y \in \mathbb{R}^d,$$

which is equivalent to

$$\phi(y) + \langle \nabla\phi(y), x - y \rangle + \frac{\sum_{i=1}^{m} w_i\mu_i}{2}\|x - y\|^2 \leq \phi(x), \qquad \forall x, y \in \mathbb{R}^d, \tag{77}$$

So, $\phi$ is $\mu$-strongly convex. □

### D.2  L-SVRP

**Lemma 11.** *Recalling that*

$$h_k := \nabla f_{\xi_k}(w_k) - \nabla f(w_k), \tag{78}$$

*we can write*

$$
\begin{aligned}
\mathrm{E}\left[\left.\langle h_k - \nabla f_{\xi_k}(x_\star), x_k - x_\star \rangle \right| x_k, w_k\right] &= \langle \mathrm{E}\left[\left.h_k - \nabla f_{\xi_k}(x_\star)\right| x_k, w_k\right], x_k - x_\star \rangle \\
&\stackrel{(78)}{=} \langle \mathrm{E}\left[\left.\nabla f_{\xi_k}(w_k) - \nabla f(w_k) - \nabla f_{\xi_k}(x_\star)\right| x_k, w_k\right], x_k - x_\star \rangle \\
&= \left\langle \underbrace{\nabla f(w_k) - \nabla f(w_k) - \nabla f(x_\star)}_{=0}, x_k - x_\star \right\rangle \\
&= 0,
\end{aligned} \tag{79}
$$

*and*

$$
\begin{aligned}
\mathrm{E}\left[\left.\|h_k - \nabla f_{\xi_k}(x_\star)\|^2 \right| x_k, w_k\right] &\stackrel{(78)}{=} \mathrm{E}\left[\left.\|\nabla f_{\xi_k}(w_k) - \nabla f(w_k) - \nabla f_{\xi_k}(x_\star)\|^2 \right| x_k, w_k\right] \\
&\stackrel{(5)}{\leq} \delta^2\|w_k - x_\star\|^2.
\end{aligned} \tag{80}
$$

**Lemma 12.** *Observe that by the way $w_{k+1}$ is defined, we have*

$$\mathrm{E}\left[\left.\|w_{k+1} - x_\star\|^2 \right| x_{k+1}, w_k\right] = p\|x_{k+1} - x_\star\|^2 + (1-p)\|w_k - x_\star\|^2. \tag{81}$$

*Taking expectation again, and applying the tower property of expectation, we get*

$$
\begin{aligned}
\mathrm{E}\left[\|w_{k+1} - x_\star\|^2\right] &= \mathrm{E}\left[\mathrm{E}\left[\left.\|w_{k+1} - x_\star\|^2 \right| x_{k+1}, w_k\right]\right] \\
&\stackrel{(81)}{=} p\mathrm{E}\left[\|x_{k+1} - x_\star\|^2\right] + (1-p)\mathrm{E}\left[\|w_k - x_\star\|^2\right],
\end{aligned} \tag{82}
$$

*which is what we set out to prove.*

# E   AUXILIARY FACTS

**Fact 1** (Every point is a fixed point). *Let $\phi : \mathbb{R}^d \to \mathbb{R}$ be a differentiable convex function. Then*

$$\text{prox}_{\gamma\phi}\left(x + \gamma\nabla\phi(x)\right) = x, \qquad \forall\gamma > 0, \quad \forall x \in \mathbb{R}^d.$$

*In particular, if $x_\star$ is a minimizer of $\phi$, then $\text{prox}_{\gamma\phi}\left(x_\star\right) = x_\star$.*

*Proof.* Pick any $x \in \mathbb{R}^d$ and $\gamma > 0$. Evaluating the proximity operator at

$$y := x + \gamma\nabla\phi(x) \tag{83}$$

gives

$$\text{prox}_{\gamma\phi}\left(y\right) = \arg\min_{x' \in \mathbb{R}^d}\left(\phi(x') + \frac{1}{2\gamma}\|x' - y\|^2\right).$$

This is a strongly convex minimization problem, and hence the (necessarily unique) minimizer

$$x' := \text{prox}_{\gamma\phi}\left(y\right) \tag{84}$$

of this problem satisfies the first-order optimality condition

$$\nabla\phi(x') + \frac{1}{\gamma}\left(x' - y\right) = 0.$$

Note that $x' = x$ satisfies this equation, and hence

$$x \overset{(84)}{=} \text{prox}_{\gamma\phi}\left(y\right) \overset{(83)}{=} \text{prox}_{\gamma\phi}\left(x + \gamma\nabla\phi(x)\right).$$

$\square$

The next statement is (Khaled and Jin, 2023, Fact 4)

**Fact 2** (Contractivity of the prox). *If $\phi$ is differentiable and $\mu$-strongly convex, then for all $\gamma > 0$ and for any $x, y \in \mathbb{R}^d$ we have*

$$\left\|\text{prox}_{\gamma\phi}\left(x\right) - \text{prox}_{\gamma\phi}\left(y\right)\right\|^2 \leq \frac{1}{(1 + \gamma\mu)^2}\|x - y\|^2.$$

*Proof.* This lemma can be seen as a tighter version of (Mishchenko et al., 2022, Lemma 5) though our proof technique is different. Note that $p(x) = \text{prox}_{\gamma h}(x)$ satisfies $\gamma\nabla h(p(x)) + [p(x) - x] = 0$, or equivalently $p(x) = x - \gamma\nabla h(p(x))$. Using this we have

$$\|p(x) - p(y)\|^2 = \|[x - \gamma\nabla h(p(x))] - [y - \gamma\nabla h(p(y))]\|^2$$
$$= \|[x - y] - \gamma\left[\nabla h(p(x)) - \nabla h(p(y))\right]\|^2$$
$$= \|x - y\|^2 + \gamma^2\|\nabla h(p(x)) - \nabla h(p(y))\|^2 - 2\gamma\left\langle x - y, \nabla h(p(x)) - \nabla h(p(y))\right\rangle. \tag{85}$$

Now note that

$$\left\langle x - y, \nabla h(p(x)) - \nabla h(p(y))\right\rangle = \left\langle p(x) + \gamma\nabla h(p(x)) - [p(y) + \gamma\nabla h(p(y))], \nabla h(p(x)) - \nabla h(p(y))\right\rangle$$
$$= \left\langle p(x) - p(y), \nabla h(p(x)) - \nabla h(p(y))\right\rangle + \gamma\|\nabla h(p(x)) - \nabla h(p(y))\|^2. \tag{86}$$

Combining Equations (85) and (86) we get

$$\|p(x) - p(y)\|^2 = \|x - y\|^2 + \gamma^2\|\nabla h(p(x)) - \nabla h(p(y))\|^2 - 2\gamma\left\langle p(x) - p(y), \nabla h(p(x)) - \nabla h(p(y))\right\rangle$$
$$- 2\gamma^2\|\nabla h(p(x)) - \nabla h(p(y))\|^2$$
$$= \|x - y\|^2 - \gamma^2\|\nabla h(p(x)) - \nabla h(p(y))\|^2 - 2\gamma\left\langle p(x) - p(y), \nabla h(p(x)) - \nabla h(p(y))\right\rangle. \tag{87}$$

Let $D_h(u,v) = h(u) - h(v) - \langle \nabla h(v), u - v \rangle$ be the Bregman divergence associated with $h$ at $u, v$. It is easy to show that

$$\langle u - v, \nabla h(u) - \nabla h(v) \rangle = D_h(u,v) + D_h(v,u).$$

This is a special case of the three-point identity (Chen and Teboulle, 1993, Lemma 3.1). Using this with $u = p(x)$ and $v = p(y)$ and plugging back into (87) we get

$$\|p(x) - p(y)\|^2 = \|x - y\|^2 - \gamma^2 \|\nabla h(p(x)) - \nabla h(p(y))\|^2 - 2\gamma \left[ D_h(p(x), p(y)) + D_h(p(y), p(x)) \right].$$

Note that because $h$ is strongly convex, we have that $D_h(p(y), p(x)) \geq \frac{\mu}{2} \|p(y) - p(x)\|^2$ and $D_h(p(x), p(y)) \geq \frac{\mu}{2} \|p(y) - p(x)\|^2$, hence

$$\|p(x) - p(y)\|^2 \leq \|x - y\|^2 - \gamma^2 \|\nabla h(p(x)) - \nabla h(p(y))\|^2 - 2\gamma\mu \|p(x) - p(y)\|^2. \tag{88}$$

Strong convexity implies that for any two points $u, v$

$$\|\nabla h(u) - \nabla h(v)\|^2 \geq \mu^2 \|u - v\|^2,$$

see (Nesterov, 2018, Theorem 2.1.10) for a proof. Using this in Equation (88) with $u = p(x)$ and $v = p(y)$ yields

$$\|p(x) - p(y)\|^2 \leq \|x - y\|^2 - \gamma^2 \mu^2 \|p(x) - p(y)\|^2 - 2\gamma\mu \|p(x) - p(y)\|^2.$$

Rearranging gives

$$\left[ 1 + \gamma^2 \mu^2 + 2\gamma\mu \right] \|p(x) - p(y)\|^2 \leq \|x - y\|^2.$$

It remains to notice that $(1 + \gamma\mu)^2 = 1 + \gamma^2\mu^2 + 2\gamma\mu$. $\qquad\square$

**Fact 3** (Recurrence). *Assume that a sequence $\{s_k\}_{k \geq 0}$ of positive real numbers for all $k \geq 0$ satisfies*

$$s_{k+1} \leq as_k + b,$$

*where $0 < a < 1$ and $b \geq 0$. Then the sequence for all $k \geq 0$ satisfies*

$$s_k \leq a^k s_0 + b \min \left\{ k, \frac{1}{1-a} \right\}. \tag{89}$$

*Proof.* Unrolling the recurrence, we get

$$s_k \leq as_{k-1} + b \leq a \left( as_{k-2} + b \right) + b \leq \cdots \leq a^k s_0 + b \sum_{i=0}^{k-1} a^i. \tag{90}$$

We can now bound the sum $\sum_{i=0}^{k-1} a^i$ in two different ways. First, since $a < 1$, we get the estimate

$$\sum_{i=0}^{k-1} a^i \leq \sum_{i=0}^{k-1} 1 = k. \tag{91}$$

Second, we sum a geometric series

$$\sum_{i=0}^{k-1} a^i \leq \sum_{i=0}^{\infty} a^i = \frac{1}{1-a}. \tag{92}$$

Note that either of these bounds can be better. So, we apply the best of these bounds. Substituting Equations (91) and (92) into (90) gives (89). $\qquad\square$

