# OpenReview forum: "A Unified Theory of Stochastic Proximal Point Methods without Smoothness"
_ICLR.cc/2025/Conference — Submitted to ICLR 2025_

### Official Review · Reviewer_AKNG · 2024-11-02

**Soundness:** 2
**Presentation:** 3
**Contribution:** 2
**Rating:** 5
**Confidence:** 4

**Summary:**

This paper presents a unified algorithmic framework of stochastic proximal point methods that encapsulates several existing SPPM-style algorithms, and some new extensions with variance reduction and arbitrary sampling as well. The proposed universal algorithm comes with a convergence analysis under relatively mild conditions, without imposing individual smoothness assumption on loss functions. A preliminary numerical study is carried out to demonstrate the performance of newly introduced algorithms on some toy linear regression tasks.

**Strengths:**

The proposed universal SPPM algorithmic framework sounds interesting, and novel as far as the reviewer knows about. The convergence analysis is generally well presented, with sufficient details provided in  a clear and neat way in the appendix sections.

**Weaknesses:**

The major concern is about the strength of the main result in Theorem 1. While exhibiting contraction rates, the bound there seems non-vanishing with respect to time instance unless $C_1=C_2=0$.  As shown in Sectons B.6-B.8 that such a requirement can be fulfilled by variance reduction variants of SPPM, but not by the standard SPPM (with $h_k \equiv 0$) as shown in Section B.2 (which is not surprising due to the fundamental limit of pure stochastic optimization methods). It will be much more interesting if the convergence bound could be made adaptive to standard SPPM with sublinear rate of convergence.

The paper also falls short on the experiment side, where only toy linear regression problems are considered with very limited data scale. Since the SPPM-type algorithms have long been acknowledged to be superior to SGD in non-smooth settings, it is more desirable to test the performances of the proposed algorithms in non-smooth learning tasks such as SVMs with hinge loss.

**Questions:**

Can the main result in Theorem 1 be strengthened to interpolate between the sublinear rate of the standard SPPMs and the linear rate of SPPMs with gradient correction or variance reduction?

---

### Official Review · Reviewer_xjpJ · 2024-11-03

**Soundness:** 2
**Presentation:** 2
**Contribution:** 2
**Rating:** 5
**Confidence:** 3

**Summary:**

The paper presents a unified theoretical framework for stochastic proximal point methods (SPPM) applied to problems that do not require smoothness assumptions but do rely on a strong convexity assumption.

The paper establishes a single convergence theorem applicable to multiple SPPM variants, providing the robust convergence behavior of these methods.

They propose a general algorithm called SPPM-LC (SPPM with Learned Correction) and integrates techniques such as variance reduction and arbitrary sampling through a flexible correction vector. Several new SPPM variants have been considered.

Experiments have been conducted to show some inherent properties of these approaches.

**Strengths:**

**S1:** This paper is generally well-written and easy to follow. It is well-grounded in theoretical analysis, establishing convergence guarantees without relying on smoothness assumptions.

**S2:** It unifies multiple SPPM variants under a single theoretical framework, making it easier to understand the relationships and convergence behavior across methods.

**S3:** The development of new SPPM variants, such as SPPM with Nonuniform Sampling and SPPM with Arbitrary Sampling, enriches the field by expanding the applicability of SPPM.

**Weaknesses:**

**W1.** This paper appears more like a review article, which may not align well with the scope of ICLR. The authors propose seven algorithms for solving differentiable convex problems. However, these algorithms may lack sufficient novelty or clear performance advantages. Readers might still be uncertain about which algorithm is the best choice.

**W2.** The framework assumes strong convexity, limiting the applicability of the results to problems that are not strongly convex or to non-convex settings.

**W3.** Algorithm 1 uses a constant, sufficiently small step size, which is rarely practical in real applications. This can introduce additional estimation errors, as highlighted in Theorem 1. In practice, a more effective approach would involve a diminishing step size [1]. The authors should at least discuss the comparison between constant and diminishing step sizes.

**W4.** The authors claim to use a weaker assumption (without smoothness); however, in their experiment, they still consider a convex least squares problem that satisfies the smoothness assumption. I think the authors should provide an example that satisfies both strong convexity and Assumption 6 but violates the $\nu$-smoothness assumption.

**W5.** The empirical experiments provided by the authors are limited to strongly convex least squares problems, which may restrict their relevance to practical applications and broader research interests.


**References:**
[1] Davis, D., & Drusvyatskiy, D. Stochastic model-based minimization of weakly convex functions. *SIOPT*, 2019.

**Questions:**

**Q1.** Is the combination of Assumption 6 and strong convexity a weaker condition than the $\nu$-smooth condition? If so, could you provide an example?

**Q2.** Could the proposed SPPM-LC framework be extended or adapted for non-convex optimization, such as weakly convex optimization?

**Q3.** How do the proposed algorithms compare with the proximal point method in [1], which is applicable to weakly convex functions using a diminishing step size?

**References:**
[1] Davis, D., & Drusvyatskiy, D. Stochastic model-based minimization of weakly convex functions. *SIOPT*, 2019.

---

### Official Review · Reviewer_y8J7 · 2024-11-04

**Soundness:** 3
**Presentation:** 3
**Contribution:** 2
**Rating:** 5
**Confidence:** 4

**Summary:**

This paper provided a framework for analyzing the stochastic proximal point methods for solving strongly convex optimization problem without smoothness. It leads to several new algorithm by introducing the techniques of sampling strategies, variance-reduction and correction.

**Strengths:**

The presentation is good. The authors provide solid theoretical analysis to support the proposed framework.

**Weaknesses:**

The motivation of this paper is unclear. The assumption without smoothness used in this paper looks not popular.

**Questions:**

My main consideration is the importance of the Assumptions 5 and 6. Although these assumptions is more general than smooth condition, it is unclear whether they are useful in real-applications. It looks that Assumptions 5 and 6 are proposed only for convergence analysis.
1. Can you provide some applications which satisfy Assumption 5 or 6 and but do not hold smoothness?
2. The experiments on linear regression look not very relevant to this paper, since it can be solved by the algorithms based on smoothness assumption.

---

### Official Review · Reviewer_y9uC · 2024-11-16

**Soundness:** 3
**Presentation:** 4
**Contribution:** 2
**Rating:** 3
**Confidence:** 5

**Summary:**

The paper deals with variance reduction techniques for stochastic proximal point algorithms, which recently emerged as a robust alternative to stochastic gradient algorithms, primarily due to their reduced sensitivity to step size. The authors develop a versatile stochastic proximal point algorithm, which can encompass both the standard stochastic proximal point algorithm, and various variance
reduced versions.  For this algorithm, and several specific instances of it,  assuming differentiability and strong convexity of the objective function, the authors derive several convergence rates for the expected objective function values, showing improvements over the standard stochastic proximal point algorithm and available theoretical results. The paper contains a section devoted to numerical experiments elucidating some theoretical aspects.

**Strengths:**

The main technical contribution of the paper is a generalization of an existing framework from functions with a Lipschitz continuous gradient to the case of differentiable functions. A non negligible contribution in my opinion is the very clear presentation with several  interesting remarks.

The assumptions and the statements are clear and the proofs technically sound.

**Weaknesses:**

The main novelty of the paper is the analysis of a unifying algorithm that allows to deal with variance reduced stochastic proximal point methods for an objective function which is  only differentiable and strongly convex.

Though the main proofs are partially different from the ones used in the related literature, the idea is not new and it is a generalization of the approach proposed in  the papers:

1) E. Gorbunov, F. Hanzely, and P. Richtarik. A unified theory of sgd: Variance reduction, sampling, quantization and coordinate descent.

2) C. Traore, V. Apidopoulos, S. Salzo, and S. Villa. Variance reduction techniques for stochastic
proximal point algorithms

So the novelty of the contribution is limited.

In general, there is a tendency in the paper to overstate the contributions, which I did not appreciate. I enclose some examples below:

1) The title claims "A unified theory of stochastic proximal poin methods without smoothness". This is misleading since the analysis is performed only for smooth (differentiable) functions, while nonsmooth ones are not covered.  With respect to the existing literature, the authors drop the assumption of Lipschitz continuity of the gradient, but in turn assume that the function is strongly convex.

2) the similarity assumption (Assumption 5) is indeed more general than the one in Khaled and Jin, but since it involves unknown quantities it is very unclear how more general it is in practice. Concrete examples of problems satisfying Assumption 5 but not the one in  Khaled and Jin must be provided in order to establish whether this generalization has some kind of impact, or not.

3) the authors often state that a unifying framework for the study of variance reduction techniques is missing, but this is false, since both the papers mentioned above provides such framework (of course, under different assumptions on the objective function). We find out that the proposed framework s is similar to the ones of the papers 1) and 2) above only on line 174, after a long discussion.

4) In the table 1:


The authors show convergence of 8 algorithms (see Table 1). However:

Algorithm 1 is the "universal" one including all the others;

SPPM* and SPPM GC are not interesting from the practical point of view since it requires the gradients at the solution

**Questions:**

1) Can you provide  class  of problems satisfying Assumption 5 but not the one in  Khaled and Jin ?
2) The experimental setting is not very interesting since it deals with a function with a LIpschitz continuous gradient (see eq 17).
Can you provide experiments in a setting covered by your assumptions but not by previous existing papers?

---

### Meta-Review · Area_Chair_WfMZ · 2024-12-21

**Metareview:**

A framework of convergence analysis is presented for a range of variations of SPPM, such as variance reduction and Point-SAGA. While the analysis is sound, reviewers assess that the novelty is limited and the contribution is a little overstated.

Experiments are limited to the simple least squares problem.

**Additional Comments On Reviewer Discussion:**

The authors did not rebut.

---

### Decision · Program_Chairs · 2025-01-22

Reject